# REGULARIZATION BY TEXTS FOR LATENT DIFFUSION INVERSE SOLVERS

**Jeongsol Kim**[*], **Geon Yeong Park**[*], **Hyungjin Chung, Jong Chul Ye**
KAIST
[*]: Equal Contribution
{jeongsol, pky3436, hj.chung, jong.ye}@kaist.ac.kr

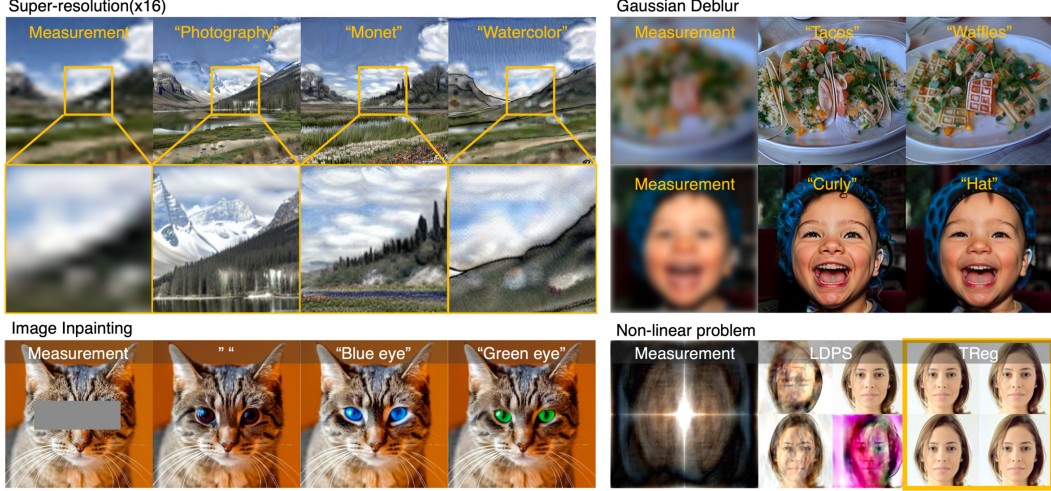

Figure 1: Representative solutions obtained by TReg for various inverse problems. TReg optimizes both data consistency *and* the semantic alignment of the solution with textual cues, by reducing the solution space with text-conditional latent regularizer. This serves as an effective semantic guidance throughout the reconstruction process.

## ABSTRACT

The recent development of diffusion models has led to significant progress in solving inverse problems by leveraging these models as powerful generative priors. However, challenges persist due to the ill-posed nature of such problems, often arising from ambiguities in measurements or intrinsic system symmetries. To address this, here we introduce a novel latent diffusion inverse solver, regularization by text (TReg), inspired by the human ability to resolve visual ambiguities through perceptual biases. TReg integrates textual descriptions of preconceptions about the solution during reverse diffusion sampling, dynamically reinforcing these descriptions through null-text optimization, which we refer to as adaptive negation. Our comprehensive experimental results demonstrate that TReg effectively mitigates ambiguity in inverse problems, improving both accuracy and efficiency.

## 1 INTRODUCTION

Consider a given forward measurement process:

$$\boldsymbol{y} = \mathcal{A}(\boldsymbol{x}) + \boldsymbol{\epsilon} \tag{1}$$

where $\mathcal{A} : \mathbb{R}^m \mapsto \mathbb{R}^n$ describes forward measurement operator, $\boldsymbol{y} \in \mathbb{R}^n$ and $\boldsymbol{x} \in \mathbb{R}^m$ represents the measurement and the true image, respectively, and $\boldsymbol{\epsilon} \in \mathbb{R}^n$ denotes the measurement noise that follows the Gaussian distribution $\mathcal{N}(0, \sigma_0^2 \mathbf{I}_n)$. Then, an inverse solver attempts to recover $\boldsymbol{x}$ from the measurement $\boldsymbol{y}$. Unfortunately, most inverse problems are ill-posed, meaning that many different visual inputs can produce identical measurements due to significant information loss during the measurement process. For example, consider Fourier phase retrieval—reconstructing

the phase information of a signal from its Fourier intensity measurements. Because of intrinsic system symmetries for shifts, rotations, and flips, it causes an input to be mapped to multiple symmetry-related outputs when inverting such systems.

Traditionally, regularization techniques have been extensively studied to address the ambiguity in inverse problems. Methods such as sparsity ($L_1$ norm), total variation (TV), and regularization by denoising (RED) have been widely employed, leveraging the statistical properties of natural images (Boyd et al., 2011; Venkatakrishnan et al., 2013; Romano et al., 2017). With the advent of diffusion models, a new class of Diffusion Inverse Solvers (DIS) (Kawar et al., 2022; Wang et al., 2022; Chung et al., 2022a; Mardani et al., 2023; Chung et al., 2023d; Rout et al., 2024) has emerged, offering superior reconstruction performance. The main idea is to leverage the diffusion model, which learns the score function of the prior distribution, to mitigate the ill-posedness of inverse problems.

While diffusion models have significantly advanced the solution of inverse problems, it is important to acknowledge the unresolved ill-posedness in certain challenging imaging systems. For instance, in Fourier phase retrieval, Diffusion Posterior Sampling (DPS) (Chung et al., 2022a; Rout et al., 2024) recover solutions more effectively than traditional methods, but it still cannot fully overcome the intrinsic symmetry, as shown in Figure 6. This limitation arises because the diffusion prior is based on image statistics, which alone may be insufficient to break the symmetry and guarantee a unique solution. A similar challenge is in systems with severely degraded measurements. Therefore, inverse problem solvers must incorporate additional cues to reduce ambiguity and fully resolve the problem.

In search of additional cues, we focus on *text* descriptions which may provide contextual information to resolve ambiguities during the sampling process, enriching the reconstructed solutions with relevant semantic content. While text conditions play a crucial role in the success of latent diffusion models, current DIS approaches based on LDM often do not fully leverage this descriptive conditioning to resolve uncertainties in recovered solutions. For instance, P2L (Chung et al., 2023d) leverages prompt embeddings as additional parameters for fine-tuning to improve alignment between solutions and measurements. However, this method primarily focuses on data consistency and thus lacks robust alignment with textual prompts. Thus, there remains a significant gap between how human perception resolves ambiguities by linguistic conditions and how DIS manage them. Bridging this gap may lead to more effective solver for addressing inverse problems.

To this end, we introduce a novel concept called Regularization by Text (TReg), implemented through latent diffusion models. TReg leverages textual descriptions that encapsulate the preconceived description of the desired solution during the reverse sampling process, framing this as a latent optimization problem to mitigate the ill-posedness. Ideally, the concepts described in the text should be exclusively reflected in the outcomes. However, we observed that manually crafted descriptions can introduce noise, leading to blurry outcomes or artifacts. To avoid this, we propose an innovative null-text optimization technique called adaptive negation, applied throughout the reverse diffusion sampling process. This method dynamically adjusts the influence of the textual guidance, ensuring that only the intended concepts align with the evolving state of the reverse sampling. As a result, TReg, when guided by a regularization prompt such as "photography of face," can break symmetries in Fourier phase retrieval and consistently produce a unique, true solution, as demonstrated in Figure 6. Furthermore, our method functions as a zero-shot inverse problem solver, robustly applicable across various domains, as illustrated in Figure 1.

## 2 BACKGROUNDS

### 2.1 LATENT DIFFUSION MODEL

Image diffusion models that operate on the pixel space are compute-heavy. So the latent diffusion model (LDM) (Rombach et al., 2022) has emerged as a class of diffusion-based generative models (Sohl-Dickstein et al., 2015; Ho et al., 2020; Song et al., 2020b), where the diffusion process is operated on low dimensional latent space instead of the pixel space. Specifically, the LDMs are first trained as a variational autoencoder by maximizing the evidence lower bound (ELBO) (Rombach et al., 2022; Kingma & Welling, 2013). Thus, the latent is represented as:

$$z = \mathcal{E}_\phi(x) := \mathcal{E}_\phi^\mu(x) + \mathcal{E}_\phi^\sigma(x) \odot \epsilon, \quad \epsilon \sim \mathcal{N}(0, \mathbf{I}), \tag{2}$$

where $\odot$ denotes the element-by-element multiplication, and $\mathcal{E}_\phi^\mu, \mathcal{E}_\phi^\sigma$ are parts of the encoder that outputs the mean and the variance of the encoder distribution. In this paper, we assume that $\mathcal{E}_\phi^\sigma(\boldsymbol{x})$ is isotropic, i.e. $\mathcal{E}_\phi^\sigma(\boldsymbol{x}) = \sigma_\mathcal{E}\mathbf{1}$. The resulting pixel domain representation can be obtained by

$$\boldsymbol{x} = \mathcal{D}_\varphi(\boldsymbol{z}) \tag{3}$$

where $\mathcal{D}_\varphi$ is the decoder.

Then, the forward diffusion is defined given the latent representation of the clean image $\boldsymbol{z}_0 = \mathcal{E}_\phi(\boldsymbol{x}) \in \mathbb{R}^d$ which perturbs $\boldsymbol{z}_0$ as $\boldsymbol{z}_t = \sqrt{\bar{\alpha}_t}\boldsymbol{z}_0 + \sqrt{1-\bar{\alpha}_t}\boldsymbol{\epsilon}$ following forward VP-SDE. Accordingly, the residual denoiser $\boldsymbol{\epsilon}_\theta(\cdot, t)$ is trained to estimate the noise $\boldsymbol{\epsilon}$ from $\boldsymbol{z}_t$ by epsilon matching:

$$\min_\theta \mathbb{E}_{\mathcal{E}_\phi(\boldsymbol{x}), \boldsymbol{\epsilon}\sim\mathcal{N}(0,\mathbf{I}),t} \left[ \|\boldsymbol{\epsilon} - \boldsymbol{\epsilon}_\theta(\boldsymbol{z}_t, t)\|_2^2 \right], \tag{4}$$

where $\boldsymbol{\epsilon}_\theta$ is often parameterized as time-conditional UNet (Ronneberger et al., 2015) in practice. Notably, the optimal solution $\boldsymbol{\epsilon}_\theta^*$ of (4) serves as an alternative approximation of a score function as $\nabla_{\boldsymbol{z}_t} \log p(\boldsymbol{z}_t) = -\boldsymbol{\epsilon}_\theta^*(\boldsymbol{z}_t, t)/t$ (Song et al., 2020b). Based on this, the generative reverse sampling from the posterior distribution $p_\theta(\boldsymbol{z}_{t-1}|\boldsymbol{z}_t, \boldsymbol{z}_0)$ can be performed as

$$\hat{\boldsymbol{z}}_{0|t} = \mathbb{E}[\boldsymbol{z}_0|\boldsymbol{z}_t] = (\boldsymbol{z}_t - \sqrt{1-\bar{\alpha}_t}\boldsymbol{\epsilon}_\theta(\boldsymbol{z}_t, t))/\sqrt{\bar{\alpha}_t} \tag{5}$$

$$\boldsymbol{z}_{t-1} = \sqrt{\bar{\alpha}_{t-1}}\hat{\boldsymbol{z}}_{0|t} + \sqrt{1-\bar{\alpha}_{t-1}}\boldsymbol{\epsilon}_\theta(\boldsymbol{z}_t, t), \tag{6}$$

which corresponds to a single iterate of deterministic DDIM sampling (Song et al., 2020a).

For a text-conditional sampling, Classifier Free Guidance (CFG) (Ho & Salimans, 2021) is widely leveraged based on the sharpened posterior distribution. Let $\varnothing$ denotes *null-text* embedding and $\boldsymbol{c}$ represents target text embedding. Then, we have $\boldsymbol{\epsilon}_\theta^\omega(\boldsymbol{z}_t, \boldsymbol{c}, t) = \boldsymbol{\epsilon}_\theta(\boldsymbol{z}_t, \varnothing, t) + \omega(\boldsymbol{\epsilon}_\theta(\boldsymbol{z}_t, \boldsymbol{c}, t) - \boldsymbol{\epsilon}_\theta(\boldsymbol{z}_t, \varnothing, t))$ where $\omega$ is a scale for the guidance. For the brevity, we will use $\boldsymbol{\epsilon}_\theta(\boldsymbol{z}, \varnothing, t) = \hat{\boldsymbol{\epsilon}}_\varnothing$, $\boldsymbol{\epsilon}_\theta(\boldsymbol{z}, \boldsymbol{c}, t) = \hat{\boldsymbol{\epsilon}}_{\boldsymbol{c}}$, and $\boldsymbol{\epsilon}_\theta^\omega(\boldsymbol{z}, \boldsymbol{c}, t) = \hat{\boldsymbol{\epsilon}}_{\boldsymbol{c}}^\omega$. For the details, please refer to Appendix A.6.

## 2.2 DIFFUSION INVERSE SOLVERS

Recently diffusion models have been emerged as powerful generative priors for inverse problems (Kawar et al., 2022; Song et al., 2022; Wang et al., 2022; Chung et al., 2022a; Mardani et al., 2023; Chung et al., 2023b). Earlier techniques in inverse imaging relied on an alternating projection method (Song et al., 2020b; Choi et al., 2021; Chung et al., 2022b), enforcing hard measurement constraints between denoising steps in either pixel or measurement spaces. More advanced strategies have been proposed to approximate the gradient of the log posterior within diffusion models, broadening the scope to tackle nonlinear problems (Chung et al., 2022a). The field has seen further expansion with methods addressing blind inverse problem (Chung et al., 2023a), 3D (Chung et al., 2023c; Lee et al., 2023), and problems of unlimited resolution (Bond-Taylor & Willcocks, 2023).

Traditionally, these methods have utilized image-domain diffusion models, but a shift has been observed towards latent diffusion models such as latent DPS (LDPS) with a fixed point of autoencoder process (Rout et al., 2024), LDPS with history update (He et al., 2023), Resample (Song et al., 2023), and leveraging prompt tuning to improve the reconstruction (Chung et al., 2024). Despite these innovations, the use of text embedding for regularization is often overlooked, missing an opportunity to fully leverage the power of multi-modal latent space—an aspect we aim to address in following.

## 3 MAIN CONTRIBUTION: REGULARIZATION BY TEXT

In this section, we present *TReg*, an iterative refinement process for solving inverse problems by fully leveraging informative text conditioning. Our framework is based on a text-conditioned latent optimization objective, which, when minimized during the reverse sampling process, progressively improves both data consistency and the alignment of the estimated solution with the provided textual cue. This regularized sampling process produces a solution that aligns to human perceptual priors. For textual alignment, our framework incorporates two key components: (**1**) text-conditional latent regularization and (**2**) adaptive negation. Specifically, the text embedding is jointly optimized to align with the current latent representation, mitigating the unintended signal components introduced by hand-crafted text descriptions. We begin by outlining the overall latent optimization framework.

## 3.1 Latent Optimization for Textual Constraint

Consider a probability density of posterior distribution $p(\boldsymbol{x}|\boldsymbol{y},\boldsymbol{z})$ where $\boldsymbol{z}$ denotes latent vector of clean image $\boldsymbol{x}$ and $\boldsymbol{y}$ denotes the measurement. By the Bayes' rule, we can see that

$$p(\boldsymbol{x}|\boldsymbol{y},\boldsymbol{z}) \propto p(\boldsymbol{y}|\boldsymbol{x},\boldsymbol{z})p(\boldsymbol{x}|\boldsymbol{z}) \propto p(\boldsymbol{z}|\boldsymbol{x},\boldsymbol{y})p(\boldsymbol{y}|\boldsymbol{x})p(\boldsymbol{x}|\boldsymbol{z}), \tag{7}$$

where we assume $p(\boldsymbol{x}|\boldsymbol{z}) := \delta(\boldsymbol{x} - \mathcal{D}_{\boldsymbol{\varphi}}(\boldsymbol{z}))$ given a well pre-trained autoencoder as in (3). Then, the objective of the maximum a posteriori (MAP) problem is defined as

$$\ell_{\text{MAP}}(\boldsymbol{z}) = -\log p(\boldsymbol{z}|\mathcal{D}_{\boldsymbol{\varphi}}(\boldsymbol{z}),\boldsymbol{y}) - \log p(\boldsymbol{y}|\mathcal{D}_{\boldsymbol{\varphi}}(\boldsymbol{z})) = \frac{\|\boldsymbol{z} - \mathcal{E}_{\phi}^{\boldsymbol{\mu}}(\mathcal{D}_{\boldsymbol{\varphi}}(\boldsymbol{z}))\|_2^2}{2\sigma_{\mathcal{E}}^2} + \frac{\|\boldsymbol{y} - \mathcal{A}(\mathcal{D}_{\boldsymbol{\varphi}}(\boldsymbol{z}))\|_2^2}{2\sigma^2}, \tag{8}$$

where the second equality holds due to the noisy measurement in (1) and VAE prior in (2).

While optimizing (8) during reverse sampling approximates posterior sampling, it does not guarantee semantic alignment between the estimated solution and the textual descriptions. To address this, we propose a text-conditional latent regularizer which drives the sampling trajectory towards clean manifold that optimally aligns with the text condition $\boldsymbol{c}$:

$$\ell_{\text{TReg}}(\boldsymbol{z}) = \|\boldsymbol{z} - \hat{\boldsymbol{z}}_{0|t}\|^2, \tag{9}$$

where the input latent $\boldsymbol{z}$ is conditioned on the text condition $\boldsymbol{c}$, and $\hat{\boldsymbol{z}}_{0|t}$ refers to a text-conditioned denoised estimate $\hat{\boldsymbol{z}}_{0|t} = \hat{\boldsymbol{z}}_{0|t}(\boldsymbol{c}) = (\boldsymbol{z}_t - \sqrt{1 - \bar{\alpha}_t}\hat{\boldsymbol{\epsilon}}_{\boldsymbol{c}}^{\omega})/\sqrt{\bar{\alpha}_t}$ derived by the Tweedie's formula (Robbins, 1992; Efron, 2011). Thus $\hat{\boldsymbol{z}}_{0|t}$ serves as a pivot, preventing the sampling trajectory from deviating significantly from the proper text-conditioned sampling path. By combining (8) and (9), the resulting proximal optimization framework is defined as follows:

$$\min_{\boldsymbol{x},\boldsymbol{z}} \quad \ell_{\text{MAP}}(\boldsymbol{z}) + \gamma\ell_{\text{Treg}}(\boldsymbol{z}) = \ell_{\text{MAP}}(\boldsymbol{z}) + \gamma\|\boldsymbol{z} - \hat{\boldsymbol{z}}_{0|t}\|^2 \tag{10}$$

$$s.t. \quad \boldsymbol{x} = \mathcal{D}_{\boldsymbol{\varphi}}(\boldsymbol{z}),$$

where we initialize the input latent $\boldsymbol{z}_0$ as $\hat{\boldsymbol{z}}_{0|t}$, the initial clean estimate of the sampling process at time $t$. While one may use off-the-shelf optimizers, we aim to solve (10) by leveraging variable splitting inspired by the alternating direction method of multipliers (Boyd et al., 2011).

Namely, using the decoder approximation and setting $\boldsymbol{x} = \mathcal{D}_{\boldsymbol{\varphi}}(\boldsymbol{z})$, the optimization problem with respect to $\boldsymbol{x}$ becomes

$$\min_{\boldsymbol{x}} \frac{\|\boldsymbol{y} - \mathcal{A}(\boldsymbol{x})\|_2^2}{2\sigma^2} + \frac{\|\boldsymbol{z} - \mathcal{E}_{\phi}^{\boldsymbol{\mu}}(\boldsymbol{x})\|_2^2}{2\sigma_{\mathcal{E}}^2} + \lambda\|\boldsymbol{x} - \mathcal{D}_{\boldsymbol{\varphi}}(\boldsymbol{z}) + \boldsymbol{\eta}\|_2^2. \tag{11}$$

Here, for simplicity, we set dual variable $\boldsymbol{\eta}$ as a zero vector and do not consider its update. Then, from the initialization with $\boldsymbol{z} = \hat{\boldsymbol{z}}_{0|t}$, we have

$$\hat{\boldsymbol{x}}_0(\boldsymbol{y}) = \arg\min_{\boldsymbol{x}} \frac{\|\boldsymbol{y} - \mathcal{A}(\boldsymbol{x})\|_2^2}{2\sigma^2} + \lambda\|\boldsymbol{x} - \mathcal{D}_{\boldsymbol{\varphi}}(\hat{\boldsymbol{z}}_{0|t})\|_2^2, \tag{12}$$

which can be solved with negligible computation cost such as conjugate gradient (CG) if $\mathcal{A}$ is a linear operation. Subsequently, using the encoder approximation and setting $\boldsymbol{z} = \mathcal{E}_{\phi}^{\boldsymbol{\mu}}(\boldsymbol{x})$ with $\boldsymbol{\eta} = \boldsymbol{0}$, the optimization problem with respect to $\boldsymbol{z}$ is derived as

$$\hat{\boldsymbol{z}}_0^{ema} = \arg\min_{\boldsymbol{z}} \zeta\|\boldsymbol{z} - \hat{\boldsymbol{z}}_0(\boldsymbol{y})\|_2^2 + \gamma\|\boldsymbol{z} - \hat{\boldsymbol{z}}_{0|t}\|^2 = \bar{\alpha}_{t-1}\hat{\boldsymbol{z}}_0(\boldsymbol{y}) + (1 - \bar{\alpha}_{t-1})\hat{\boldsymbol{z}}_{0|t}, \tag{13}$$

where the second equality is a closed-form solution with the initialization of $\hat{\boldsymbol{z}}_0(\boldsymbol{y}) := \mathcal{E}_{\phi}(\hat{\boldsymbol{x}}_0(\boldsymbol{y}))$. Here, $\zeta, \gamma$ are empirically chosen to satisfy $\bar{\alpha}_{t-1} = \zeta/(\zeta + \gamma)$ to hold the second equality of (13). Specifically, we set the interpolation coefficient to prioritize $\hat{\boldsymbol{z}}_0(y)$ during the final phase of reverse sampling, ensuring fine-grained refinement for improved data consistency. Meanwhile, the text-conditioned pivot relatively dominates the early stages of the sampling process, establishing the coarse layout and improving text adherence. Note that the estimated solution $\hat{\boldsymbol{z}}_0^{ema}$ is obtained by the interpolation between initial text-conditioned estimate $\hat{\boldsymbol{z}}_{0|t}$ and $\hat{\boldsymbol{z}}_0(\boldsymbol{y})$ derived with the measurement $\boldsymbol{y}$. Finally, by integrating the updated latent $\hat{\boldsymbol{z}}_0^{ema}$, a single iterate of the resulting DDIM sampling (Song et al., 2020a) at $t$ reads:

$$\boldsymbol{z}'_{t-1} = \sqrt{\bar{\alpha}_{t-1}}\hat{\boldsymbol{z}}_0^{ema} + \sqrt{1 - \bar{\alpha}_{t-1}}\tilde{\boldsymbol{\epsilon}}_t, \tag{14}$$

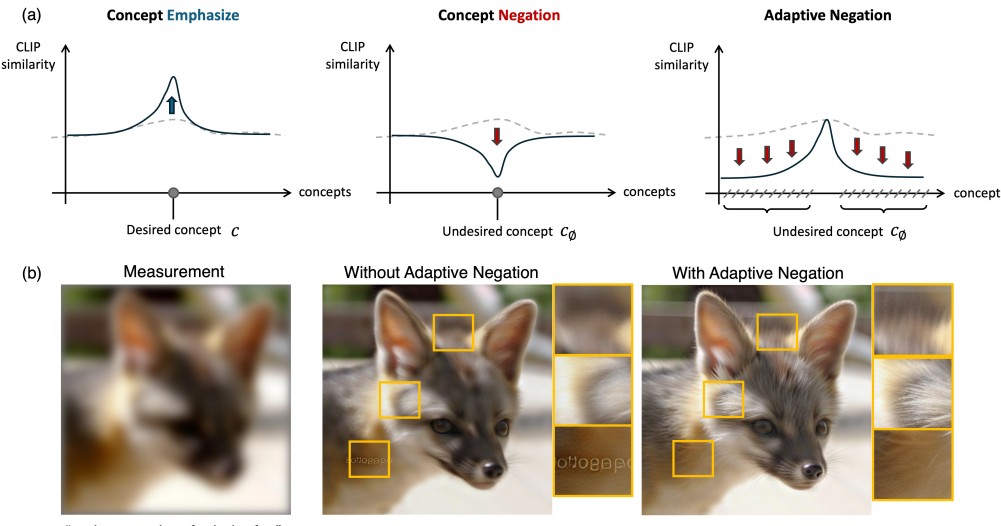

Figure 2: (a) Concept of adaptive negation. Compared to concept emphasize and negation which targets a specific concept, the adaptive negation tries to suppress concepts except the desired one. (b) Adaptive negation is crucial to avoid artifacts on reconstruction.

where $\tilde{\epsilon}_t$ denotes the total noise given by

$$\tilde{\epsilon}_t := \frac{\sqrt{1 - \bar{\alpha}_{t-1} - \eta^2 \tilde{\beta}_t^2} \hat{\epsilon}_c^\omega + \eta \tilde{\beta}_t \epsilon}{\sqrt{1 - \bar{\alpha}_{t-1}}}. \tag{15}$$

In (14), $\eta \in [0, 1]$ is a parameter controlling the stochasticity of the update rule. Empirically, we find that $\eta \tilde{\beta}_t = \sqrt{\bar{\alpha}_{t-1}} \sqrt{1 - \bar{\alpha}_{t-1}}$ results in robust performance. In other words, the total noise is computed by $\tilde{\epsilon}_t := \sqrt{1 - \bar{\alpha}_{t-1}} \hat{\epsilon}_\theta + \sqrt{\bar{\alpha}_{t-1}} \epsilon$. For the convergence analysis, please refer to the Appendix B.

## 3.2 ADAPTIVE NEGATION FOR TEXTUAL CONSTRAINT

The primary component of the proposed optimization framework includes $\hat{z}_{0|t}$, a pivot for sampling paths conditioned on text embeddings, which effectively narrows the solution space. While conditioning on a well-chosen embedding is crucial for semantic alignment, relying on error-prone, hand-crafted prompts can result in suboptimal reconstructions, as illustrated in Figure 2(b).

To mitigate this, we propose the joint optimization of the text embedding during the reverse sampling process in an adaptive manner based on image representations. Specifically, we employ concept negation (Ho & Salimans, 2021) to suppress invalid concepts which relatively enhances the intended semantic contents, as shown in Figure 2(a). We prioritize concept negation over concept emphasis via direct prompt tuning (Lester et al., 2021), as the latter may disrupt the original intended semantic contents embedded in the text prompt.

Here, the null-text embedding $c_\varnothing$ in CFG is regarded as a representation of concepts to be suppressed. The goal of adaptive negation is to update $c_\varnothing$ to exclude concepts that is already captured by the latent $z$ through latent optimization. During the reverse sampling, we update $c_\varnothing$ to minimize the following similarity in the CLIP (Radford et al., 2021) embedding space:

$$\ell_\varnothing(c_\varnothing) = \langle \mathcal{T}_{img}(\hat{x}_0(y)), c_\varnothing \rangle, \tag{16}$$

where $\mathcal{T}_{img}$ denotes CLIP image encoder, and $\hat{x}_0(y)$ denotes clean estimates on pixel space which is the solution of (12). By optimizing (16), $c_\varnothing$ is "adaptively" updated to encode concepts that are not prominent in the image representations, thereby allowing it to represent complementary concepts. The computational cost of this optimization is negligible as shown in Table 4, since it only involves the optimization of $c_\varnothing$ without backpropagating through the denoiser.

## 3.3 LATENT DPS WITH UPDATED NULL-TEXT

Based on the defined proximal optimization on clean latent domain, we found that alternating latent DPS steps further improves the data consistency, especially for the image inpainting and phase retrieval problems. In this case, we obtain the standard LDPS gradient with null text embedding $c_\varnothing$, and update the intermediate noisy sample:

$$z_{t-1} = z'_{t-1} - \rho_t \nabla_{z_t} \|\mathcal{A}(\mathcal{D}_{\varphi}(\hat{z}_{0|t}(\hat{c}_\varnothing))) - y\|^2 \tag{17}$$

where $\rho_t$ denotes the step size that weights the likelihood, similar to (Chung et al., 2022a; Rout et al., 2024). In image super-resolution and deblurring, we bypass the DPS update for computational efficiency. Nevertheless, our algorithm can effectively solve the problem and shows comparable performance with DPS update as shown in Figure 7. To sum up, the proposed algorithm is described as in Algorithm 1.

## 4 EXPERIMENTS

The goal of text regularization (TReg) is to refine the solution space and reduce ambiguity by leveraging text prompts as additional cues. We assume that both the measurement and a text prompt describing the solution are provided for the inverse problem. To highlight the effectiveness of ambiguity reduction, we evaluate TReg under extreme measurement conditions, in contrast to other inverse problem solvers. Our evaluation focuses on two key aspects: (1) the effectiveness of TReg in resolving ambiguity through text, and (2) the accuracy of the resulting solution, which includes alignment with both the text and the measurements. For further details on the experimental settings, please refer to the Appendix.

### 4.1 EXPERIMENTAL SETTINGS

**Forward models.** For linear inverse problems, we select bicubic super-resolution with scale factor 16, Gaussian deblur with kernel size 61 and sigma 5.0, and box inpainting where the masked reason is designed to encompasses the eyes and mouth

---

**Algorithm 1** Inverse problem solving with TReg

**Require:** Pretrained LDM $\epsilon_\theta$, Null text embedding $c_\varnothing$, Text embedding $c$, VAE encoder $\mathcal{E}_\phi$, VAE decoder $\mathcal{D}_\varphi$, Forward operation $\mathcal{A}$, Measurement $y$, CLIP image encoder $\mathcal{T}_{img}$, Update Range $\Gamma$ [a]
**Initialization** $z_T \sim \mathcal{N}(0, \mathbf{I})$
**for** $t \in [T, 1]$ **do**
  $\hat{z}_{0|t} = (z_t - \sqrt{1-\bar{\alpha}_t}\hat{\epsilon}_c^\omega)/\sqrt{\bar{\alpha}_t}$
  $\tilde{\epsilon}_t \leftarrow$ Compute noise using (15)
  **if** $t \in \Gamma$ **then**
    // Latent Optimization
    $\hat{x}_0 \leftarrow \mathcal{D}(\hat{z}_{0|t})$
    $\hat{x}_0(y) \leftarrow$ Equation (12)
    $\hat{z}_0(y) \leftarrow \mathcal{E}(\hat{x}_0(y))$
    $\hat{z}_0^{ema} \leftarrow \bar{\alpha}_{t-1}\hat{z}_0(y) + (1-\bar{\alpha}_{t-1})\hat{z}_{0|t}$
    $z_{t-1} \leftarrow \sqrt{\bar{\alpha}_{t-1}}\hat{z}_0^{ema} + \sqrt{1-\bar{\alpha}_{t-1}}\tilde{\epsilon}_t$
    // Adaptive Negation
    $c_\varnothing \leftarrow c_\varnothing - \eta\nabla_\varnothing \text{sim}(\mathcal{T}_{img}(\hat{x}_0(y)), c_\varnothing)$
  **else**
    $z_{t-1} \leftarrow \sqrt{\bar{\alpha}_{t-1}}\hat{z}_{0|t} + \sqrt{1-\bar{\alpha}_{t-1}}\tilde{\epsilon}_t$
  **end if**
**end for**

---

[a]Please refer to Appendix A.4.

of either animal or human. The forward operators are defined by following Kawar et al. (2022); Wang et al. (2022); Song et al. (2022); Chung et al. (2023b). For non-linear inverse problems, we choose Fourier phase retrieval and gamma correction. For details of the forward operators, please refer to Appendix. For all tasks, we add measurement noise that follows the Gaussian distribution with zero mean and noise scale $\sigma_0^2 = 0.01$.

**Dataset and text prompt.** We prepare measurement-text sets where the text describes solution. The simplest way is to leverage the ground-truth class label as a text description. However, to further explore the capability of *TReg* reducing solution space according to given text description, we provide perceptually estimated (non-ground-truth) classes as text cues. Here, we should carefully set proper class for measurement to avoid ignoring the provided guidance.[1]

We decide to use Food-101 dataset (Bossard et al., 2014) for quantitative evaluation since it contains complex patterns, leading to high ambiguity. Also, given that all images fall under the 'food' category, it is more straightforward to select an appropriate $c$ aligned with the original caption. Specifically, we

---

[1]For example, *"dog"* for the measurement generated by baby image cannot effectively reduce solution space.

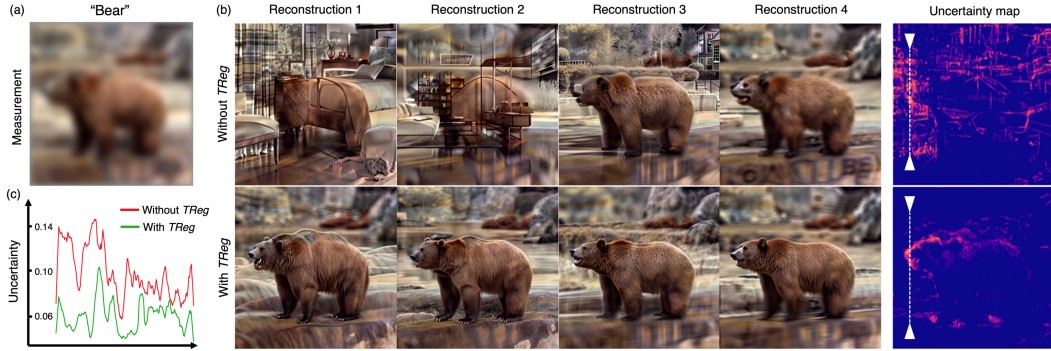

Figure 3: TReg effectively reduce ambiguity of solution with text-based regularization. (a) Given measurement and text description. (b) Multiple reconstructions and pixel-wise variance without and with text regularization. (c) Variance measured over white dotted line on uncertainty map.

leverage 250 images from each of the "fried rice" and "ice cream" classes. For the case that given text description is different from the original classes, we use *"spaghetti"* for "fried rice" and *"macarons"* for "ice cream". For the qualitative comparison, we additionally use a validation set of ImageNet, AFHQ, FFHQ and LHQ datasets. Used text prompt is provided with reconstructed samples.

**Baselines.** As the first attempt to solve inverse problems using text-driven regularization, we establish baselines by combining 1) data consistency-preserving methods and 2) text-guided editing methods. Notably, these baselines involve sequential processing, while the proposed method stands out for its efficiency, eliminating the need for pre/post-hoc techniques. For data consistency methods, we utilize the measurement itself and PSLD (Rout et al., 2024) which is an LDM-based inverse problem solver. Regarding text-guided editing methods, we opt for state-of-the-art techniques, namely Delta Denoising Score (DDS) (Hertz et al., 2023) and Plug-and-Play diffusion (PnP) (Tumanyan et al., 2023). Since PnP involves an inversion process, we handle it as a two-stage approach. Also, we use PSLD with text guidance as our baseline by computing estimated noise via CFG. For the inpainting task, we utilize Stable-Inpaint, a fine-tuned stable diffusion model for the inpainting task, and Repaint (Lugmayr et al., 2022) as our baseline. Finally, in the Fourier phase retrieval task, we compared the results with and without the text prompt using our framework to emphasize the effect of text guidance in breaking symmetry. In other words, for the baseline, we provided null text for the text condition.

## 4.2 EXPERIMENTAL RESULTS

**Ambiguity reduction.** To assess the effectiveness of TReg in reducing ambiguity in reconstruction through text regularization, we quantify pixel-level variance across multiple reconstructions from blurry measurements with and without text descriptions (Figure 3).[2] Specifically, for the reconstruction without text description, we use a null-text $c_\varnothing$ as a text description for the solution. All other components such as latent optimization (Section 3.1) and adaptive negation (Section 3.2) are retained.

As shown in Figure 3(b), TReg leads to consistent solutions corresponding to the given text description, while reconstructed images exhibit multiple solutions, such as a car in the background or a bedroom, in the absence of text regularizaiton. This discrepancy is clearly observed in pixel-level variance in Figure 3(c). While solutions with TReg are successfully refined, some residual uncertainties are observed, especially at head positioning or mouth shape. However, it is noteworthy that those residual uncertainties do not violate the data consistency and text description, and TReg is effective in resolving ambiguity via text.

**Accuracy of obtained solution: use true class as $c$.** TReg is an inverse problem solver that uses text description of solution to refine the solution space. To evaluate the accuracy of obtained solution via TReg, we first prepare set of measurement-text pairs where the original class of measurement is given as text. For both SR and Gaussian deblur tasks, we report PSNR and FID of reconstructions

---

[2]Specifically, we repeat reconstruction 10 times with different random seeds for a single measurement. The true image is sampled from ImageNet validation set.

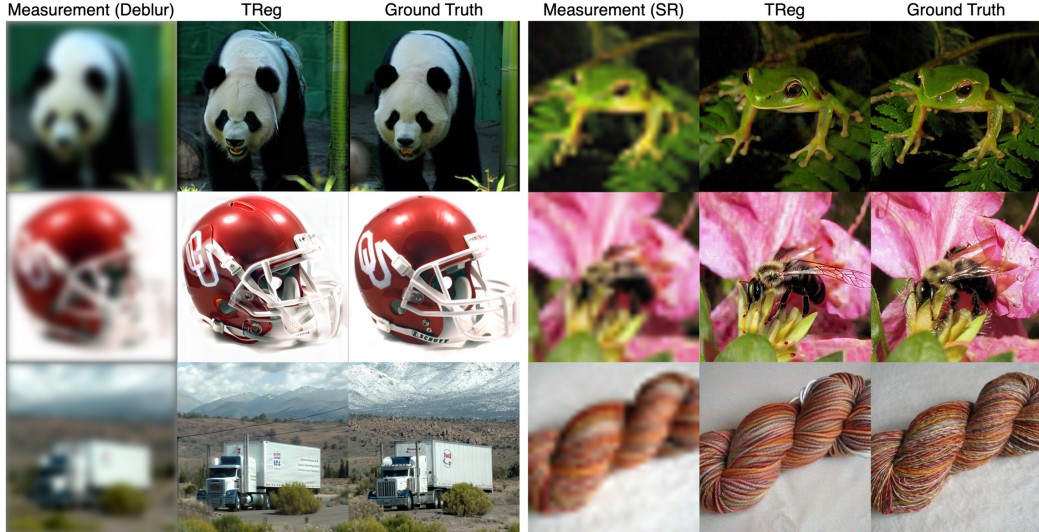

| Measurement (Deblur) | TReg | Ground Truth | Measurement (SR) | TReg | Ground Truth |

Figure 4: Reconstruction by TReg where the original class is given as text description: "A photo of <class>."

| Class: Ice-Cream | SRx16 | | Deblur | |
|---|---|---|---|---|
| Method | PSNR ↑ | FID ↓ | PSNR ↑ | FID ↓ |
| DDRM | 16.98 | 268.5 | 17.41 | 223.2 |
| PGDM | 14.95 | 260.5 | 13.87 | 295.8 |
| PSLD | 17.73 | 280.1 | 21.89 | 213.6 |
| PSLD + CFG | 16.47 | 176.1 | 20.76 | 192.9 |
| Our (w/o AN) | **22.05** | 144.4 | **23.91** | 131.2 |
| Our (w/ AN) | 21.09 | **124.2** | 23.73 | **120.6** |

| Class: Fried-rice | SRx16 | | Deblur | |
|---|---|---|---|---|
| Method | PSNR ↑ | FID ↓ | PSNR ↑ | FID ↓ |
| DDRM | 16.71 | 244.0 | 17.08 | 195.3 |
| PGDM | 14.58 | 246.6 | 14.01 | 276.2 |
| PSLD | 17.32 | 320.3 | 21.09 | 187.3 |
| PSLD + CFG | 14.97 | 150.8 | 20.49 | 168.3 |
| Our (w/o AN) | **21.26** | 132.0 | **22.99** | 133.4 |
| Our (w/ AN) | 19.98 | **97.40** | 22.83 | **112.1** |

Table 1: Quantitative result for text "ice cream".   Table 2: Quantitative result for text "fried rice".

| # stages | Method | Fried Rice → Spaghetti (*SRx16*) | | | Ice Cream → Macaron (*SRx16*) | | | Fried Rice → Spaghetti (*Deblur*) | | | Ice Cream → Macaron (*Deblur*) | | |
|---|---|---|---|---|---|---|---|---|---|---|---|---|---|
| | | LPIPS ↓ | CLIP-sim ↑ | y-MSE ↓ | LPIPS ↓ | CLIP-sim ↑ | y-MSE ↓ | LPIPS ↓ | CLIP-sim ↑ | y-MSE ↓ | LPIPS ↓ | CLIP-sim ↑ | y-MSE ↓ |
| 1 | Ours | 0.769 | **0.303** | 0.005 | 0.771 | 0.314 | **0.004** | **0.737** | 0.300 | 0.013 | **0.743** | 0.312 | **0.011** |
| | P2L | **0.756** | 0.265 | 0.020 | **0.743** | 0.293 | 0.020 | 0.798 | 0.274 | **0.013** | 0.750 | 0.252 | **0.011** |
| 2 | PSLD | **0.756** | 0.265 | 0.020 | **0.743** | 0.293 | 0.020 | 0.798 | 0.274 | **0.013** | 0.750 | 0.252 | **0.011** |
| | PnP | 0.826 | 0.251 | **0.005** | 0.808 | 0.239 | 0.005 | 0.798 | 0.259 | 0.015 | 0.752 | 0.248 | **0.011** |
| | PSLD+DDS | 0.788 | 0.247 | 0.010 | 0.772 | **0.328** | 0.013 | 0.768 | 0.247 | 0.014 | 0.752 | 0.300 | 0.015 |
| 3 | PSLD+PnP | 0.801 | 0.291 | 0.014 | 0.784 | 0.306 | 0.008 | 0.761 | **0.312** | 0.016 | 0.751 | **0.319** | 0.013 |

Table 3: Quantitative evaluation of SRx16 and Gaussian Deblurring task. Mean values are reported.
**Bold**: the best score, underline: the second best.

in Table 1 and 2. Evaluation demonstrates the effectiveness of proposed method compared to other diffusion-based inverse solvers. The proposed solver without adaptive negation tends to achieve higher PSNR than with adaptive negation, since it obtains blurry images with missing details. Thus, FID score is improved with adaptive negation which is aligned with qualitative comparison in Figure 2. Also, TReg effectively reconstruct solutions for other dataset such as ImageNet (Figure 4) which implies the robustness of the proposed solver to various image domain based on powerful diffusion prior. For more analysis, please refer to Appendix section G.

**Accuracy of obtained solution: use different class as $c$.**

Using TReg, we can find solution of inverse problem according to text prompt. To examine this property, we solve the aforementioned problems where the given text is different with the original class of the measurement. The proper solution should satisfy 1) data consistency with measurement and 2) alignment with given prompt. We evaluate the quality of obtained solution based on three metrics: LPIPS, the mean squared error on measurement domain, namely y-MSE[3], and the CLIP similarity between reconstruction and given text prompt following the prior works in image editing (Tumanyan et al., 2023; Hertz et al., 2023; Park et al., 2024). Note that there is no ground truth in this case, so we do not evaluate PSNR or SSIM.

---
[3]This is equivalent to data consistency loss, $\|\boldsymbol{y} - \mathcal{A}(\boldsymbol{x})\|_2^2$.

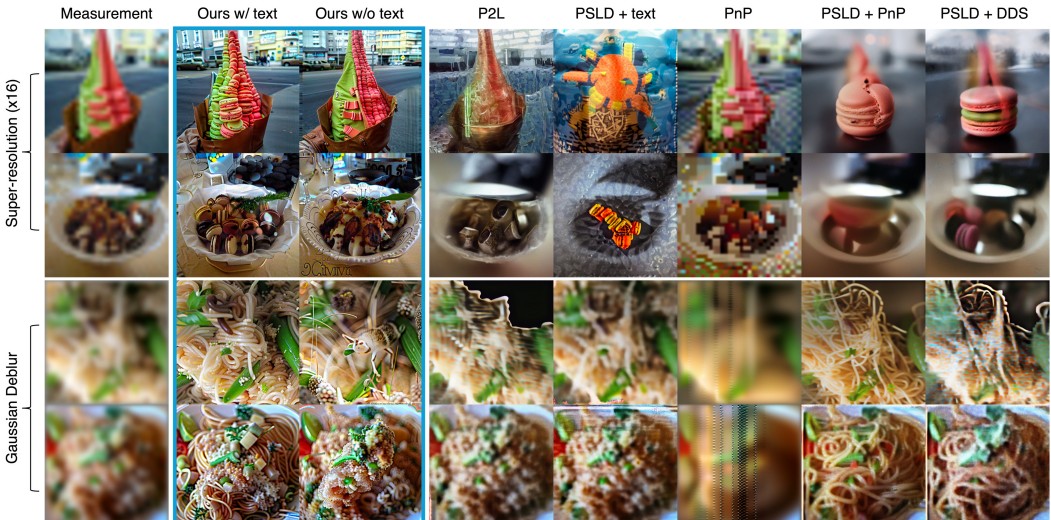

Figure 5: Reconstructions when given text prompt differs from the original class.

Table 3 shows that TReg achieves superior performance compared to baselines. Specifically, TReg find solution with high quality while preserving data consistency, as evidenced by lower LPIPS and y-MSE values. For some cases, PSLD gives lower LPIPS but solutions lose data consistency and exhibit large y-MSE values, which is also shown in the fourth column of Figure 5. TReg is comparable to or outperforms baseline methods, emphasizing that TReg is better than a straightforward application of image editing algorithms in solving text-guided inverse problems. For more analysis, please refer to Appendix section H.

Direct application of the editing algorithm to the measurement, referred to as PnP, yields sub-optimal performance across all metrics. In the case of y-MSE, PnP achieves a lower value due to minimal alterations made to the measurement, as depicted in the last column of Figure 5. Also, the performance of sequential approaches shows that reconstruction error is accumulated at each stage, resulting in substantial errors in the final outcome. In contrast, TReg effectively integrates the given text prompt while preserving robust data consistency, thereby validating its efficacy. Meanwhile, we also compare the results of our method without text prompt. When text is replaced by null-text (third column in Figure 5), it provides a solution that aligns well with the given measurement but generates arbitrary structures. This result highlights the impact of TReg that reduces the ambiguity of inverse problem solving. PSLD with CFG guidance (fourth column in Figure 5) also fails to appropriately reflect given text prompt, as there is no consideration for text-guidance in solving inverse problems.

### 4.3 ADDITIONAL RESULTS ON NON-LINEAR INVERSE PROBLEMS

Although we propose solving latent optimization (10) using variable splitting for computational efficiency, any off-the-shelf optimizer could be employed to address, for example, non-linear inverse problems. We demonstrate in Figure 6 that TReg improves the accuracy of solutions for non-linear problems through text regularization. Specifically, the results for Fourier Phase Retrieval emphasize that current diffusion-based inverse problem solvers, such as Latent DPS, are still insufficient for breaking the symmetry caused by the loss of phase information. In contrast, text regularization via TReg successfully breaks this symmetry and consistently produce a unique solution across multiple trials. Additionally, for non-uniform deblurring with gamma correction, TReg enhances reconstruction quality compared to Latent DPS. Notably, text conditioning through CFG in the latent DPS introduces unintended noisy structures, which are absent when using TReg.

## 5 RELATED WORK

Leveraging text prompts for inverse problem solving has been studied in several recent works. TextIR (Bai et al., 2023) incorporates text embedding from CLIP during the training of an image restoration network to constrain the solution space in the case of extreme inverse problems. DA-

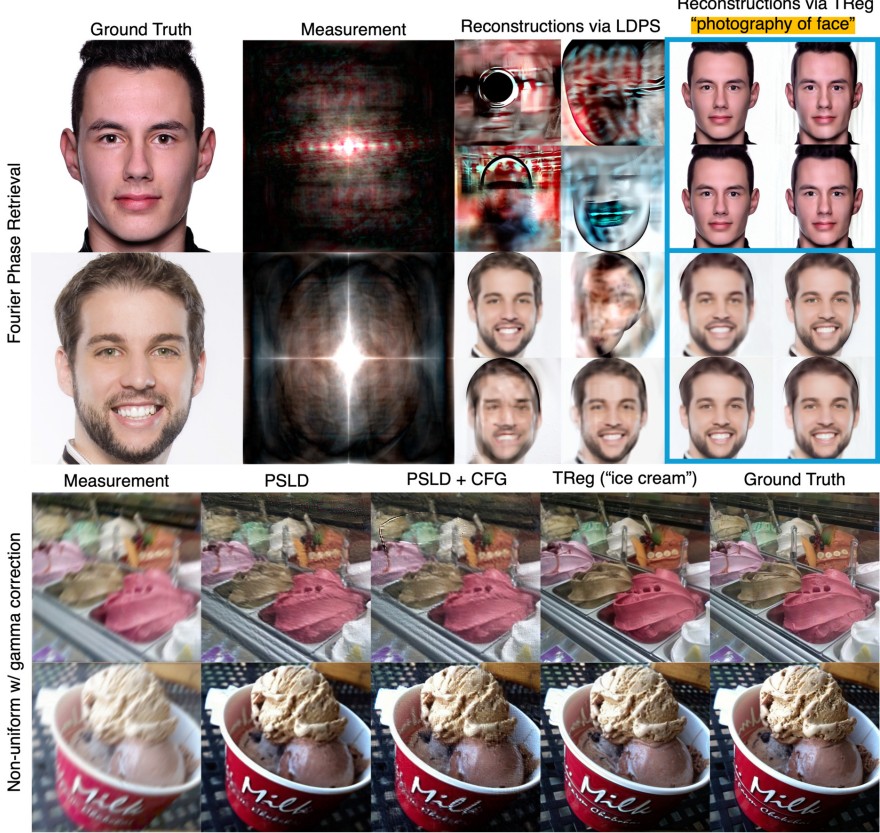

Figure 6: Reconstructions for non-linear inverse problems: Fourier Phase retrieval and Non-uniform deblurring with gamma correction.

CLIP (Luo et al., 2023) takes a similar approach of leveraging the embeddings and applies it to diffusion bridges. P2L (Chung et al., 2023d) proposes to automatically tune text embedding on-the-fly while running a generic diffusion inverse problem solver (More discussions in Appendix C). Notably, the latter two methods (Luo et al., 2023; Chung et al., 2023d) are focused on improving the overall performance rather than constraining the solution space. TextIR has similar objective to *TReg*, but requires task-specific training. In this regard, to the best of our knowledge, *TReg* is the first general diffusion-based inverse problem solver that does not require task-specific training, while being able to incorporate text conditions to effectively control the solution space in general inverse problems.

# 6 CONCLUSION

This study introduced a novel concept of latent diffusion inverse solver with *regularization by texts*, namely *TReg*. The proposed solver minimizes ambiguity in solving inverse problems, effectively reducing uncertainty and improving accuracy in visual reconstructions, bridging the gap between human perception and machine-based interpretation. To achieve this, we derived LDM-based reverse sampling steps to minimize data consistency with text-driven regularization, consisting of adaptive negation. Specifically, to effectively integrate textual cues and guide reverse sampling, the paper introduced a null text optimization approach. Experimental results and visualizations demonstrated that text-driven regularization effectively reduces the uncertainty of solving inverse problems, and further enables text-guided control of signal reconstruction.

## ACKNOWLEDGMENTS

This work was supported by the National Research Foundation of Korea under Grant RS-2024-00336454 and by the Institute for Information & Communications Technology Planning & Evaluation

(IITP) grant funded by the Korea government (MSIT) (RS-2019-11190075, Artificial Intelligence Graduate School Program, KAIST).

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

# A  IMPLEMENTATION DETAILS

In this section, we provide further details on implementation of *TReg*. The code will be available to public on `https://github.com/TReg-inverse/Treg`.

## A.1  FORWARD OPERATIONS

**Box inpainting** In the case of the box inpainting task, our intention is to employ a universal mask that encompasses the eyes and mouth of either animal or human. For this, we generate a rectangular mask based on averaged images across all data points within each dataset.

**Fourier phase retrieval** For Fourier phase retrieval, we obtain the Fourier magnitude by applying the Fourier transform to the image. In the formula, the forward model is expressed as $\boldsymbol{y} \sim \mathcal{N}(\boldsymbol{y}||FP\boldsymbol{x}_0|, \sigma_0^2\mathbf{I})$, where $F$ denotes the 2D Discrete Fourier Transform (DFT), $P$ represents oversampling, that is implemented by adding 256 zero-padding to each side, and $\mathbf{x}_0$ denotes the clean image.

## A.2  STABLE-DIFFUSION AND CLIP

We leverage the pre-trained Latent Diffusion Model (LDM), including an auto-encoder and the U-net model, provided by diffusers. Due to the limitation of its modularity, we implement the sampling process for inverse problems by ourselves, rather than changing given pipelines. The Stable-diffusion v1.5 is utilized for every experiment in this work, and the ViT-L/14 backbone and its checkpoint is used for CLIP image encoder for adaptive negation.

## A.3  CONJUGATE GRADIENT METHOD

In this work, we apply the CG method to find a solution to the following problem:

$$\min_{x} \frac{\|\boldsymbol{y} - \mathcal{A}\boldsymbol{x}\|_2^2}{2\sigma^2} + \lambda \|\boldsymbol{x} - \mathcal{D}_{\varphi}(\hat{\boldsymbol{z}}_{0|t})\|_2^2. \tag{18}$$

As the objective function is a convex function, the solution $\boldsymbol{x}^*$ should satisfy

$$-\mathcal{A}^{\top}(\boldsymbol{y} - \mathcal{A}\boldsymbol{x}^*) + \lambda(\boldsymbol{x}^* - \mathcal{D}_{\varphi}(\hat{\boldsymbol{z}}_{0|t})) = 0, \tag{19}$$

where the coefficients are absorbed to $\lambda$. Then, we can formulate it as a linear system as

$$(\lambda \mathbf{I} + \mathcal{A}^{\top}\mathcal{A})\boldsymbol{x}^* = \lambda \mathcal{D}_{\varphi}(\hat{\boldsymbol{z}}_{0|t}) + \mathcal{A}^{\top}\boldsymbol{y} \tag{20}$$

where $\mathbf{A} = \lambda \mathbf{I} + \mathcal{A}^{\top}\mathcal{A}$ and $\mathbf{b} = \lambda \mathcal{D}_{\varphi}(\hat{\boldsymbol{z}}_{0|t}) + \mathcal{A}^{\top}\boldsymbol{y}$. Thus, we can solve (20) by CG method. In this work, we use 5 iterations of CG update with $\lambda = 1e - 4$ for each time step if not explicitly stated otherwise.

## A.4  CG UPDATE RANGE

The data consistency update with CG algorithms is applied for a subset of sampling steps as described by $\Gamma$ in Algorithm 1 of the main paper. Our empirical observation show that this partial data consistency update achieves better trade-off between the image reconstruction consistency and the latent stability. In the absence of the data consistency update, we employ the deterministic DDIM sampling step by setting $\eta\tilde{\beta}_t = 0$ or apply DPS gradient to improve the reconstruction quality. Overall, for inverse problems except the Fourier phase retrieval, we set the Network Function Evaluation (NFE) to 200 and design $\Gamma = \{t|t \bmod 3 = 0, t \leq 850\}^4$, where mod denotes the modulo operation. For the Fourier phase retrieval, we set $\Gamma = \{t|t \bmod 10 = 0\}$.

---

[4]The pre-trained Stable-diffusion uses $T = 1000$. Thus, $t = 850$ is equivalent to the sampling step with NFE = 170.

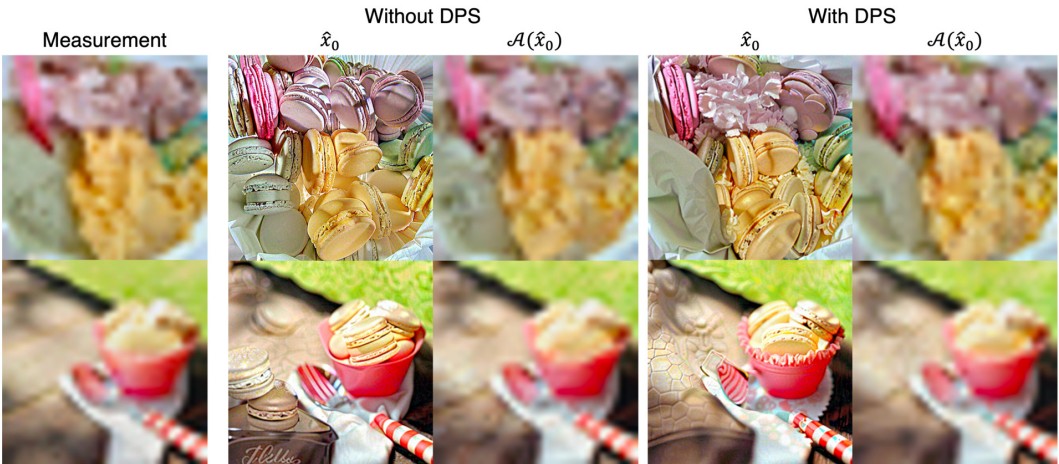

Figure 7: Regardless of DPS guidance, we can obtain proper solution through TReg for super-resolution and deblurring. Original class: "ice cream", Given text prompt: "macaron". $\hat{x}_0$ represents obtained solution and $\mathcal{A}(\hat{x}_0)$ denotes simulated measurement with $\hat{x}_0$.

---

**Algorithm 2** Inverse problem solving with TReg with DPS

---

**Require:** $\epsilon\theta, \varnothing, c, \mathcal{E}, \mathcal{D}, \mathcal{A}, y, \mathcal{T}_{img}$
  $z_T \sim \mathcal{N}(0, \mathbf{I})$
  **for** $t \in [T, 1]$ **do**
    **if** $t \in \Gamma$ **then**
      $\hat{z}_{0|t} = (z_t - \sqrt{1 - \bar{\alpha}_t}\hat{\epsilon}_c^{\omega_1}(z_t))/\sqrt{\bar{\alpha}_t}$
      $\tilde{\epsilon}_t \leftarrow$ Compute noise using (15)
      $\hat{x}_0 \leftarrow \mathcal{D}(\hat{z}_{0|t})$
      $\hat{x}_0(y) \leftarrow$ Equation (12)
      $\hat{z}_0(y) \leftarrow \mathcal{E}(\hat{x}_0(y))$
      $\hat{z}_0^{ema} \leftarrow \bar{\alpha}_{t-1}\hat{z}_0(y) + (1 - \bar{\alpha}_{t-1})\hat{z}_{0|t}$                            ▷ Latent Optimization
      $z_{t-1} \leftarrow \sqrt{\bar{\alpha}_{t-1}}\hat{z}_0^{ema} + \sqrt{1 - \bar{\alpha}_{t-1}}\tilde{\epsilon}_t$
      $c_\varnothing \leftarrow c_\varnothing - \eta\nabla_\varnothing \text{sim}(\mathcal{T}_{img}(\hat{x}_0(y)), c_\varnothing)$               ▷ Adaptive negation
    **else**
      $\hat{z}_{0|t} = (z_t - \sqrt{1 - \bar{\alpha}_t}\hat{\epsilon}_c^{\omega_2}(z_t))/\sqrt{\bar{\alpha}_t}$
      $\tilde{\epsilon}_t \leftarrow$ Compute noise using (15)
      $z'_{t-1} \leftarrow \sqrt{\bar{\alpha}_{t-1}}\hat{z}_{0|t} + \sqrt{1 - \bar{\alpha}_{t-1}}\tilde{\epsilon}_t$
      $z_{t-1} \leftarrow z'_{t-1} - \rho_t\nabla_{z_t}\|\mathcal{A}(\mathcal{D}_\varphi(\hat{z}_{0|t})) - y\|$
    **end if**
  **end for**

---

### A.5 TREG WITH DPS UPDATE

In the main manuscript, we only describe the pseudocode of TReg without DPS update for the readability. However, the algorithm could readily conduct an additional DPS update as follow. The pseudocode could be described as in Algorithm 2. We have used this additional DPS step for the box inpainting and Fourier phase retrieval task with step size $\rho_t = \sqrt{\bar{\alpha}_{t-1}}$. For box inpainting task, we set CFG scale $\omega_2$ for DPS gradient to 0, while we set CFG scale $\omega_2 = \omega_1$ in Fourier phase retrieval task. For other tasks, we found that reconstructions quality is promising even without DPS as shown in Figure 7.

### A.6 CFG GUIDANCE SCALE

The classifier-free guidance (CFG) is defined as

$$\hat{\epsilon}_\theta = \epsilon_\theta(z_t, \phi, t) + \omega(\epsilon_\theta(z_t, c, t) - \epsilon_\theta(z_t, \phi, t)) \tag{21}$$

where $\omega$ is a scale for the guidance. In other words, $\omega$ could be interpreted as a magnitude for the negation. Thus, we leverage $\omega$ to control the extent of enhancement applied to a given text prompt. In fact, we can regulate the same feature by adjusting the null-text update strategy, including null-text update frequency and number of optimization iterations per time step. However, the CFG guidance scale is simpler and more intuitive than adjusting the null-text update strategy. Hence, we have used a proper CFG scale while fixing the null-text update schedule $\Gamma$.

For the main experiments with the Food101 dataset in the main paper, we use the default scale 7.5 for all results. For the Fourier phase retrieval problem, we set the CFG scale $\omega_1 = \omega_2 = 4.0$. For other results, CFG scale among $\{3.0, 4.0, 5.0, 7.5\}$ provide robust performance for various datasets including FFHQ, AFHQ, LHQ (landscape dataset), and ImageNet.

### A.7 OPTIMIZATION PROBLEM FOR FOURIER PHASE RETRIEVAL

In contrast other inverse problems with linear operation, Fourier phase retrieval involves non-linear operation so we cannot leverage CG to solve the optimization problem in pixel space. Thus, we use Adam optimizer with learning rate $1e-3$ and $\beta_1 = 0.9, \beta_2 = 0.999$ to obtain the solution of

$$\min_{\boldsymbol{x}} \frac{\|\boldsymbol{y} - \mathcal{A}(\boldsymbol{x})\|_2^2}{2\sigma^2} + \lambda \|\boldsymbol{x} - \hat{\boldsymbol{x}}_0\|_2^2 \tag{22}$$

by setting $\lambda = 0$. This induces additional computational costs to solve the optimization problem. However, as analyzed in Section D, these costs are comparable to those of other baseline algorithms, while consistently achieving the reconstruction of a unique solution.

## B CONVERGENCE ANALYSIS

In section 3.1, we optimize the full latent proximal objective by leveraging alternate variable splitting, where the data consistency (12) is optimized in pixel space and the text-conditional proximal optimization (13) is mainly solved in latent space. Here, we analyze the convergence of the proposed refinement process for a better understanding.

A direct theoretical convergence analysis is challenging due to the complexity arising from the transition between pixel and latent spaces. However, we provide an informative empirical analysis by evaluating the progression of two objectives during the reverse sampling process: (**a**) data consistency measured in pixel space and (**b**) diffusion training objective (also known as the denoising score matching loss) with a given text prompt.

In formula, we plot (**a**) $\|\boldsymbol{y} - A(\hat{\boldsymbol{x}}_{0|t})\|^2$ and (**b**) $\|\boldsymbol{\epsilon} - \boldsymbol{\epsilon}_\theta(\boldsymbol{z}_t, \boldsymbol{c}, t)\|^2$ as in (24). The first objective indicates how well the solution aligns with the given measurement. The second objective represents how well the solution follows the marginal distribution of the pretrained diffusion model, conditioned on the given text. We present results for two cases as the main paper: where the given text prompt matches the true label and where it differs from the true label. For both cases, we

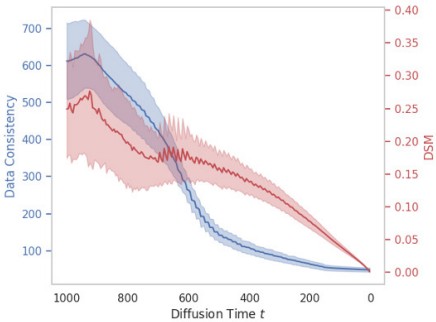

Figure 8: Data consistency and DSM loss during reconstruction. Mean $\pm$ Std for 100 samples are plotted.

plot the mean and standard deviation of these objectives, computed across 100 samples. Figure 8 and 9 shows that both objectives decrease significantly during the reverse sampling process, demonstrating that TReg empirically promotes convergence. Notably, the data consistency of TReg improves particularly in the later sampling phase, which is in line with the interpolative constant setting ($\bar{\alpha}_{t-1} = \zeta/(\zeta + \gamma)$) in (13).

## C COMPARISON TO P2L

Here, we compare TReg with P2L (Chung et al., 2023d) in detail and highlight the key differences. As mentioned in Sec. 1, we note that P2L is *orthogonal* to our approach, despite their apparent

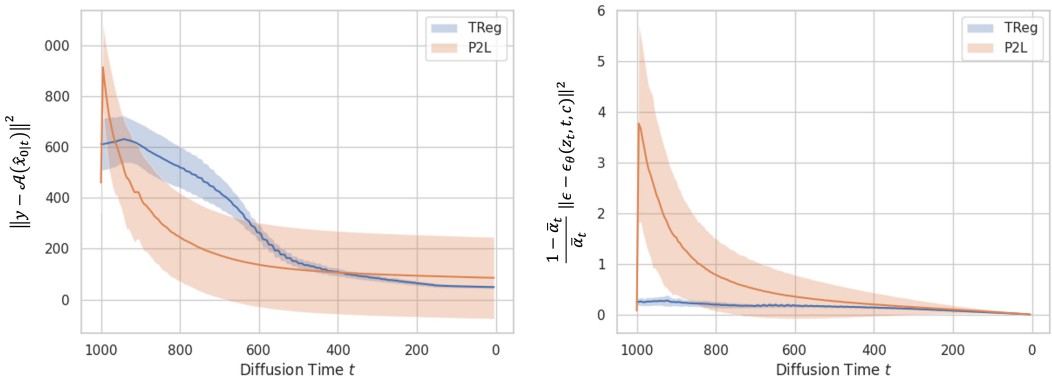

Figure 9: Data consistency and Text-conditioned score matching loss for TReg and P2L. Mean $\pm$ Std for 100 reconstructions are plotted. Task: Gaussian Deblurring. Original Class: "Risotto". Target Text: "French Fries". P2L initializes the prompt embedding with the target text ("French Fries").

similarities. Specifically, P2L treats the (null-text) prompt as an additional parameter to improve the data consistency, optimizing it as:

$$\mathcal{C}^*_{t,\text{P2L}} = \arg\min_{\mathcal{C}} \|\boldsymbol{y} - \mathcal{A}(\mathcal{D}_{\boldsymbol{\varphi}}(\hat{\boldsymbol{z}}_{0|t}(\mathcal{C})))\|_2^2, \tag{23}$$

which corresponds to eq.(12) of Chung et al. (2023d).

Unlike TReg, P2L does not guide outputs toward a specific mode aligned with semantic linguistic conditions. It lacks support for Classifier-Free Guidance (CFG) and relies solely on conditional scores. While both methods leverage prompt embeddings, they do so in fundamentally different ways: P2L focuses on data consistency, whereas TReg reduces the solution space by incorporating rich perceptual estimates of noisy measurements in the form of linguistic descriptions.

Several empirical analysis verify these distinctions (see Sec. G). First, while P2L initialize $\boldsymbol{c}$ as a null text embedding in general, for the comparison, we *adapt* P2L as a target text-based baseline by providing a ground-truth textual descriptions. Quantitatively, TReg outperforms P2L even under these ideal scenarios (see Sec. G for more discussions). We remark that P2L requires *twice* the neural function evaluations (NFEs) compared to TReg, as it recomputes Tweedie estimates after each prompt update (refer to Table 4 and Sec. D for details).

Fig. 9 further highlights the fundamental gap between TReg and P2L. Specifically, we monitor the normalized text-conditional score matching loss (i.e. score matching distillation loss (Poole et al., 2023)) throughout the reverse sampling for both TReg and P2L:

$$\ell(\boldsymbol{z}) = \frac{1 - \bar{\alpha}_t}{\bar{\alpha}_t} \|\boldsymbol{\epsilon} - \boldsymbol{\epsilon}_\theta(\sqrt{\bar{\alpha}_t}\boldsymbol{z} + \sqrt{1 - \bar{\alpha}_t}\boldsymbol{\epsilon}, \boldsymbol{c}, t)\|^2 \tag{24}$$

$$= \frac{1 - \bar{\alpha}_t}{\bar{\alpha}_t} \|\boldsymbol{\epsilon} - \boldsymbol{\epsilon}_\theta(\boldsymbol{z}_t, \boldsymbol{c}, t)\|^2. \tag{25}$$

This evaluates how well the intermediate solutions progressively align with the clean data manifold *and* the corresponding text condition $\boldsymbol{c}$. As shown in Fig. 9 (right), TReg consistently achieves better progressive alignment with textual conditions, particularly during the early stages of sampling. These early stages are critical for the overall spatial layout, eventually establishing semantic textual alignment. Overall, TReg demonstrates qualitative and quantitative advantages over P2L by achieving more efficient sampling, better data consistency, and textual alignment.

## D RUNTIME ANALYSIS

We propose text regularization by formulating a proximal optimization problem in the latent space. Thus, additional time to solve the optimization problem is required compared to reverse diffusion process. Notably, TReg does not require excessive computation time although we introduce a two-step optimization approach. This is because we employ an accelerated optimization method, like

|  | PSLD | PSLD w/ AN | P2L | TReg |
|---|---|---|---|---|
| Linear | 1.47 | 1.47 | 2.00 | 0.26 |
| Non-linear | 1.12 | 1.12 | – | 1.09 |

Table 4: Wall-clock runtime (min) of each algorithm for NFE=200.

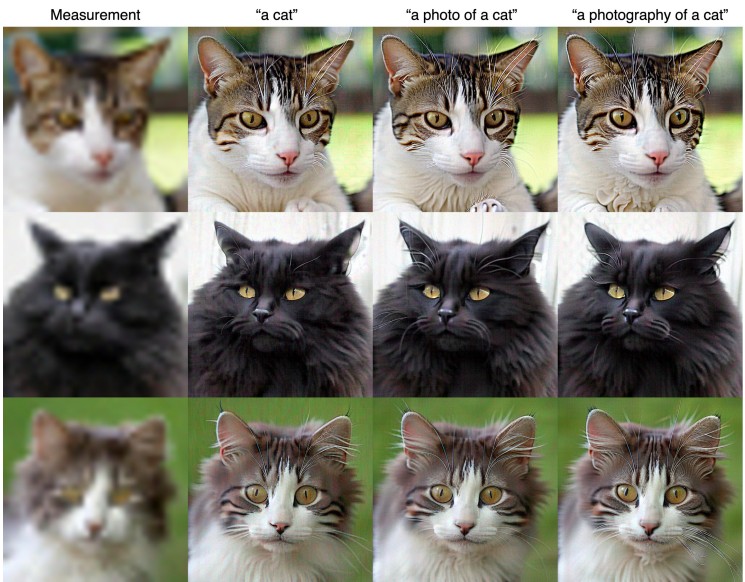

Figure 10: Reconstruction by TReg with various text prompts. Super-resolution (x16) task on AFHQ-cat.

conjugate gradietn (CG), and derive a closed-form solution for subsequent problem. As a result, TReg demonstrates superior efficiency compared to the baseline algorithm, as shown in Table 4. To evaluate the efficiency, we measure the runtime for a Gaussian deblurring task and Fourier phase retrieval with PSLD and TReg, as representative tasks for linear and non-linear inverse problem. Specifically, we did not use the fixed-point constraint of PSLD for the non-linear inverse problem, as $\mathcal{A}^\top$ is not properly defined in this case. Table 4 shows that TReg requires significantly shorter time to obtain solutions for linear inverse problem. Also, for the non-linear problem where the off-the-shelf optimizer is leveraged, TReg is faster than PSLD since the optimizaion variable is clean latent estimate so it does not require back-propagation through denoising UNet. Furthermore, the optimization variable for adaptive negation is limited to the text embedding, which incurs minimal additional computational cost. Consequently, the runtime for adaptive negation is negligible.

## E  ABLATION STUDY ON TEXT PROMPT

We conduct an ablation study on text prompts to assess whether the reconstructions vary based on the given text. Specifically, for the AFHQ-cat validation set, we examine the solutions for a super-resolution task obtained by TReg using the prompts "a cat," "a photo of a cat", and "a photography of a cat." As shown in Figure 10, the reconstructions are not highly sensitive to grammatical variations in the text prompts.

## F  ROBUSTNESS TO JPEG ARTIFACTS

A practical scenario for inverse problems involves measurements that include additional noise beyond Gaussian noise, such as artifacts introduced by JPEG compression. In this section, we evaluate the robustness of TReg to JPEG artifacts. Specifically, we apply a JPEG compression algorithm[5] to

---

[5]We use the forward operation defined by DDRM (Kawar et al., 2022), `https://github.com/bahjat-kawar/ddrm-jpeg`.

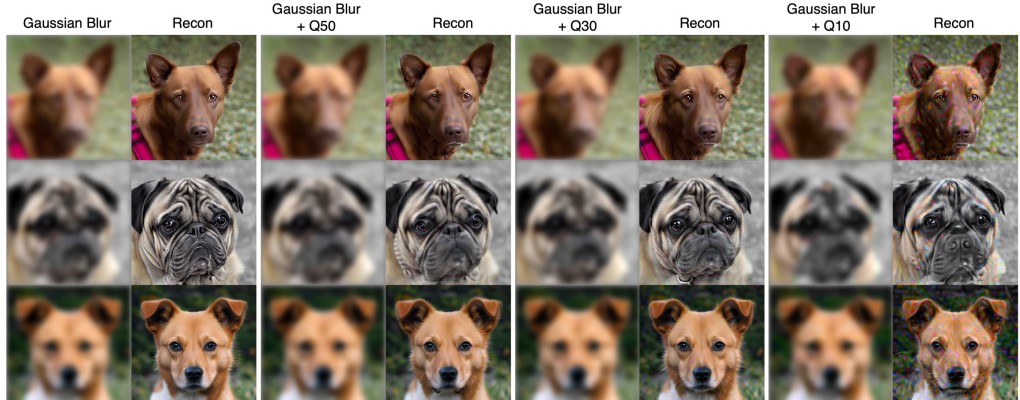

Figure 11: Reconstructions in the presence of JPEG compression artifacts applied after Gaussian blur on AFHQ-dog validation set.

| Method | FFHQ (SRx16) | | FFHQ (Deblur) | | FFHQ (Inpaint) | | AFHQ (SRx16) | | AFHQ (Deblur) | | AFHQ (Inpaint) | |
|---|---|---|---|---|---|---|---|---|---|---|---|---|
| | PSNR ↑ | FID ↓ | PSNR ↑ | FID ↓ | PSNR ↑ | FID ↓ | PSNR ↑ | FID ↓ | PSNR ↑ | FID ↓ | PSNR ↑ | FID ↓ |
| PSLD | 20.01 | 142.8 | **24.82** | 59.41 | 19.76 | **60.97** | 16.48 | 113.4 | 20.52 | 125.5 | 16.93 | 104.7 |
| PSLD + CFG | 15.67 | 146.4 | 22.61 | 109.4 | 16.82 | 90.72 | 16.45 | 123.8 | 20.58 | 125.3 | 15.17 | 130.3 |
| P2L | 21.94 | **72.02** | 23.03 | 91.15 | 16.84 | 85.32 | **19.99** | 121.7 | 20.96 | 85.80 | 16.07 | 138.4 |
| TReg | **22.60** | 82.71 | **24.82** | **40.24** | **19.95** | 66.93 | 19.60 | **37.13** | **21.13** | **35.47** | **17.39** | **51.97** |

Table 5: Quantitative evaluation on FFHQ and AFHQ validation sets. The ground-truth class labels are given as text descriptions. **Bold** represents the best and underline denotes the second best.

the measurement and then add Gaussian noise. For JPEG compression, we vary the quality factor (QF) at 50, 30, and 10, where a higher quality factor indicates less compression. It is important to note that we do not include JPEG compression in our forward operation, as the purpose of this study is to assess TReg's robustness to measurement noise. For the prompt, we use "a photo of a dog". As shown in Figure 11, TReg demonstrates robustness up to Q30, providing nearly identical reconstructions. At Q10, however, significant noise is introduced into the measurement, resulting in degraded reconstructions. For such extreme JPEG artifacts, incorporating JPEG compression into the forward operation could be considered.

## G    ADDITIONAL EVALUATION WITH GROUND-TRUTH CLASSES

The main experiments focus on practical scenarios where perceptually estimated descriptions simulate real-world user-provided cues for better reconstruction. In this section, for completeness, we also evaluate an additional scenario with 'ground-truth' textual descriptions of the noisy measurements. Using benchmarks like FFHQ, AFHQ-cat, and AFHQ-dog, where ground-truth class labels (e.g., 'dog', 'cat') are available, P2L is adapted as a text-based baseline by initializing prompt embeddings with these ground-truth label descriptions.

Table 5 and 6 shows that TReg outperforms other baselines, including the computationally expensive P2L. Fig. 10, 11, 12, and 13 further illustrate the qualitative advances. These results highlight that TReg utilizes textual information more effectively and could achieve even better performance with reliable descriptive inferences of noisy measurements.

| Method | ImageNet (SRx16) | | ImageNet (Deblur) | | ImageNet (Inpaint) | |
|---|---|---|---|---|---|---|
| | PSNR ↑ | FID ↓ | PSNR ↑ | FID ↓ | PSNR ↑ | FID ↓ |
| PSLD | 18.04 | 170.5 | **20.97** | 115.9 | 17.41 | 90.13 |
| P2L | 18.62 | 141.1 | 19.58 | 117.0 | 15.94 | 119.3 |
| TReg | **19.71** | **69.65** | 20.66 | **55.92** | **18.11** | **50.67** |

Table 6: Quantitative evaluation on ImageNet 1k validation sets. The ground-truth class labels are given as text descriptions. **Bold** represents the best and underline denotes the second best.

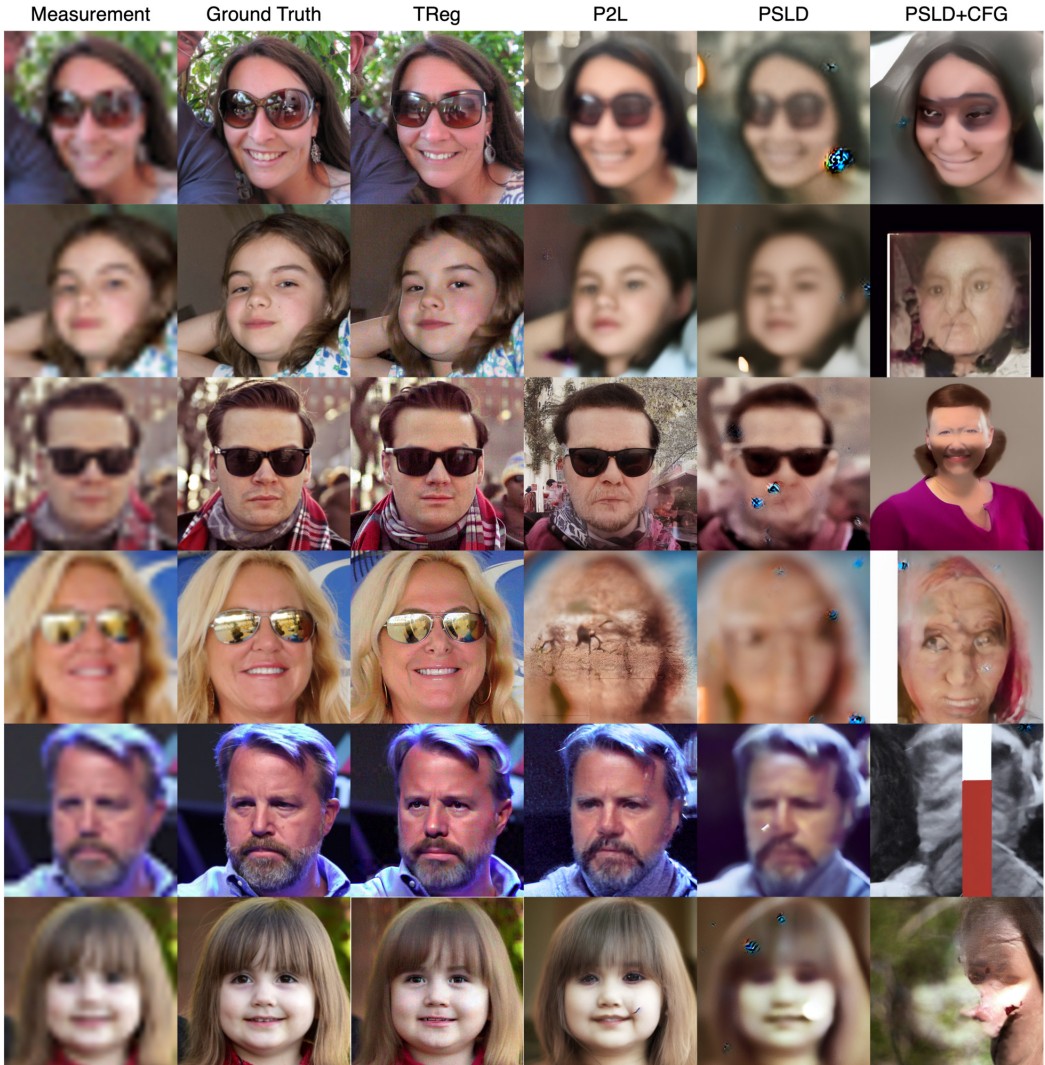

| Measurement | Ground Truth | TReg | P2L | PSLD | PSLD+CFG |

Figure 12: Qualitative comparison for super-resolution (x16) task on FFHQ

# H    ADDITIONAL EVALUATION WITH ESTIMATED CLASSES

To further demonstrate whether TReg can reliably and generally reconstruct "perceptually plausible" solutions from a single noisy measurement with text description, we conduct additional evaluation on ImageNet validation set which contains 1000 classes.

Specifically, we construct a validation set from ImageNet comprising image-text pairs where the text description differs from the original class. First, we constructed a 1k ImageNet validation set encompassing all ImageNet classes, following the approach used in P2L. For each validation sample, the ground-truth class is known. Next, to identify a "perceptually plausible" text description, we measured pairwise LPIPS for all noisy measurements simulated from the 1k ImageNet validation set and selected the target text description corresponding to the class label with the lowest LPIPS samples. This process resulted in a 1k ImageNet validation set with both original and target classes (e.g. "great white shark" and "albatross", see Figure 16). Finally, we solved the inverse problem on this validation set using the experimental setup outlined in the main experiment.

Table 6 demonstrates that TReg effectively reconstructs solutions aligned with the given target class while satisfying data consistency, as evidenced by improved FID, CLIP similarity scores, and yMSE. In contrast, baseline methods exhibit significant trade-offs among these metrics. For example, PSLD achieves the best FID in the deblurring task, but its reconstructions fail to align with the

| Measurement | Ground Truth | TReg | P2L | PSLD | PSLD+CFG |
|---|---|---|---|---|---|

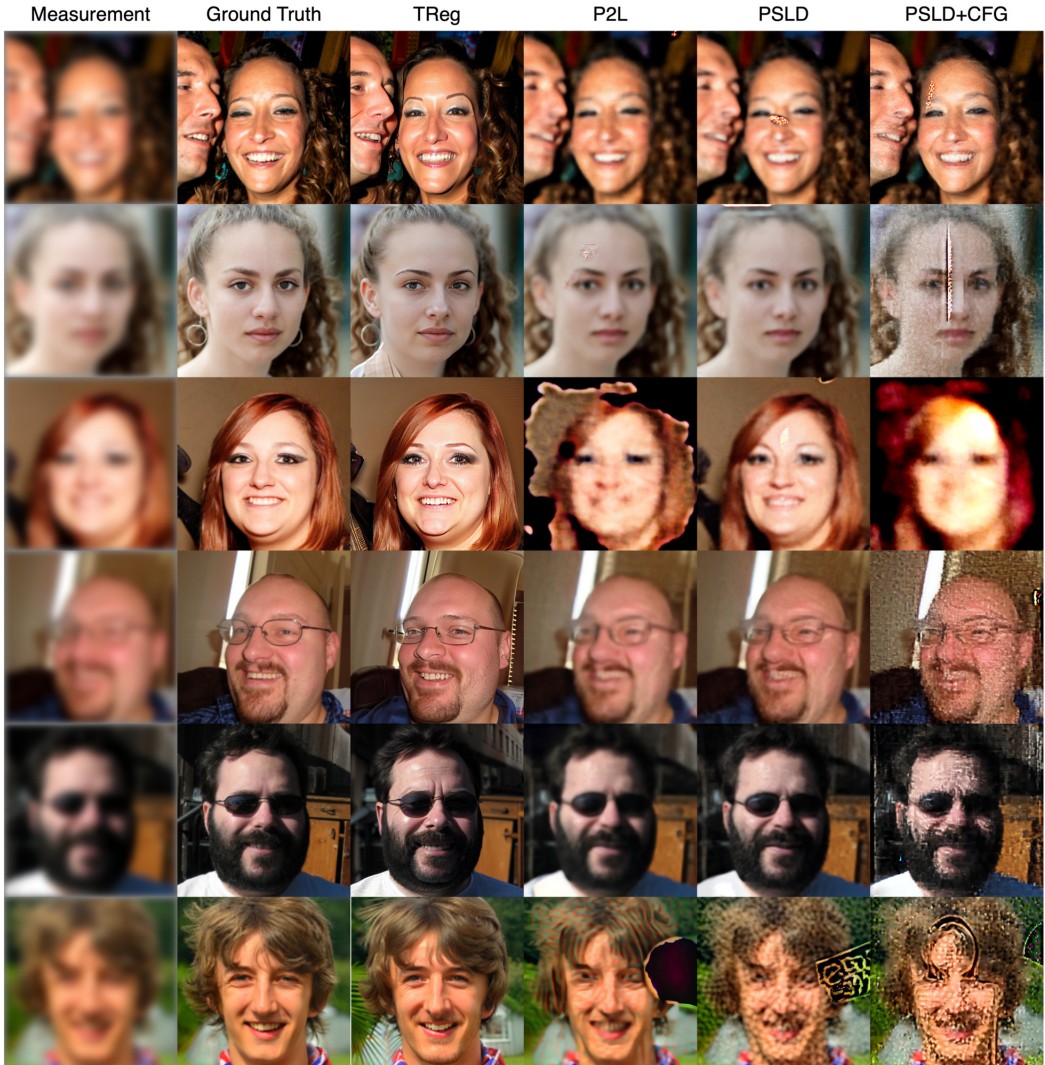

Figure 13: Qualitative comparison for deblurring (Gauss) task on FFHQ

given text description, as indicated by low CLIP similarity. Similarly, PSLD + CFG in the SR task achieves higher CLIP similarity but fails to maintain data consistency. In comparison, TReg delivers better or comparable performance across all metrics. These results highlight the versatility of text regularization with TReg in solving inverse problems for a wide range of natural images, extending beyond categories like person, cat, and dog.

# I   ADDITIONAL RESULTS

## I.1   TEXT-GUIDED BOX INPAINTING

We conduct a qualitative comparison for box inpainting task. As illustrated in Figure 17 (a), TReg reconstructs image properly according to the given text prompt with higher fidelity compared to baseline methods. From Figure 17 (b), we repeatedly observe that text regularization enables the discovery of solutions as intended. It allows the discovery of even rare solutions, such as a cat with green eyes. Additionally, in Figure 17 (c), TReg demonstrates its capability to solve the inpainting problem by composing concepts. In other words, TReg can narrow down the solution space by utilizing not only a singular concept but also multiple concepts simultaneously.

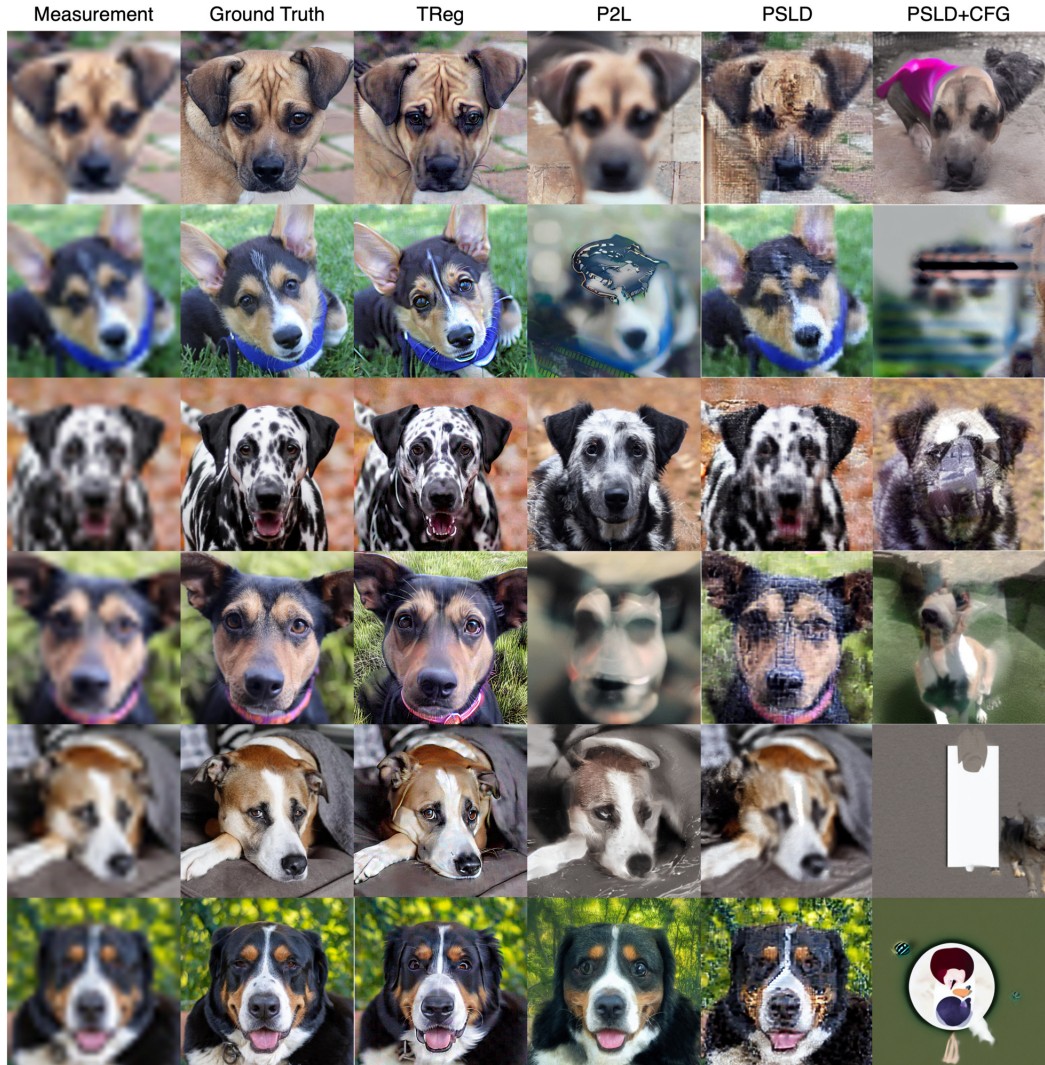

Figure 14: Qualitative comparison for super-resolution (x16) task on AFHQ

Furthermore, we provide uncurated reconstruction results in Figure 18 to demonstrate the robustness of the TReg. We provide two box inpainting scenarios: a rectangular inpainting task on the AFHQ-dog validation set with the text prompt "a photo of a dog with glasses", and a square inpainting task on the AFHQ-cat validation set with the text prompt "a photo of a dog". For both cases, TReg successfully fills out the masked regions according to provided text descriptions.

## I.2   SUPER-RESOLUTION AND DEBLURRING

TReg is a zero-shot inverse problem solver based on diffusion prior trained on large scale of text-image dataset, implying its robustness to diverse set of texts and image domain. Thus, we further investigate the robustness of TReg in reducing solution space based on given text prompt. Figure 19 illustrates reconstructions with various text prompts. Regardless of data domain, TReg effectively finds the solution by preserving data consistency and reflecting text prompt. Furthermore, the result directly shows the existence of multiple solutions that satisfied given forward model, which is an evidence of the ill-posedness nature of the inverse problem. In addition, we examine the reconstruction for inverse problem defiend on FFHQ dataset with text prompts "baby face" and "adult face". As shown in Figure 20, the proposed method successfully reconstructs images according to the given prompt, while the baseline method (sequential approach of PSLD and DDS) shows inferior reconstruction quality.

| Measurement | Ground Truth | TReg | P2L | PSLD | PSLD+CFG |

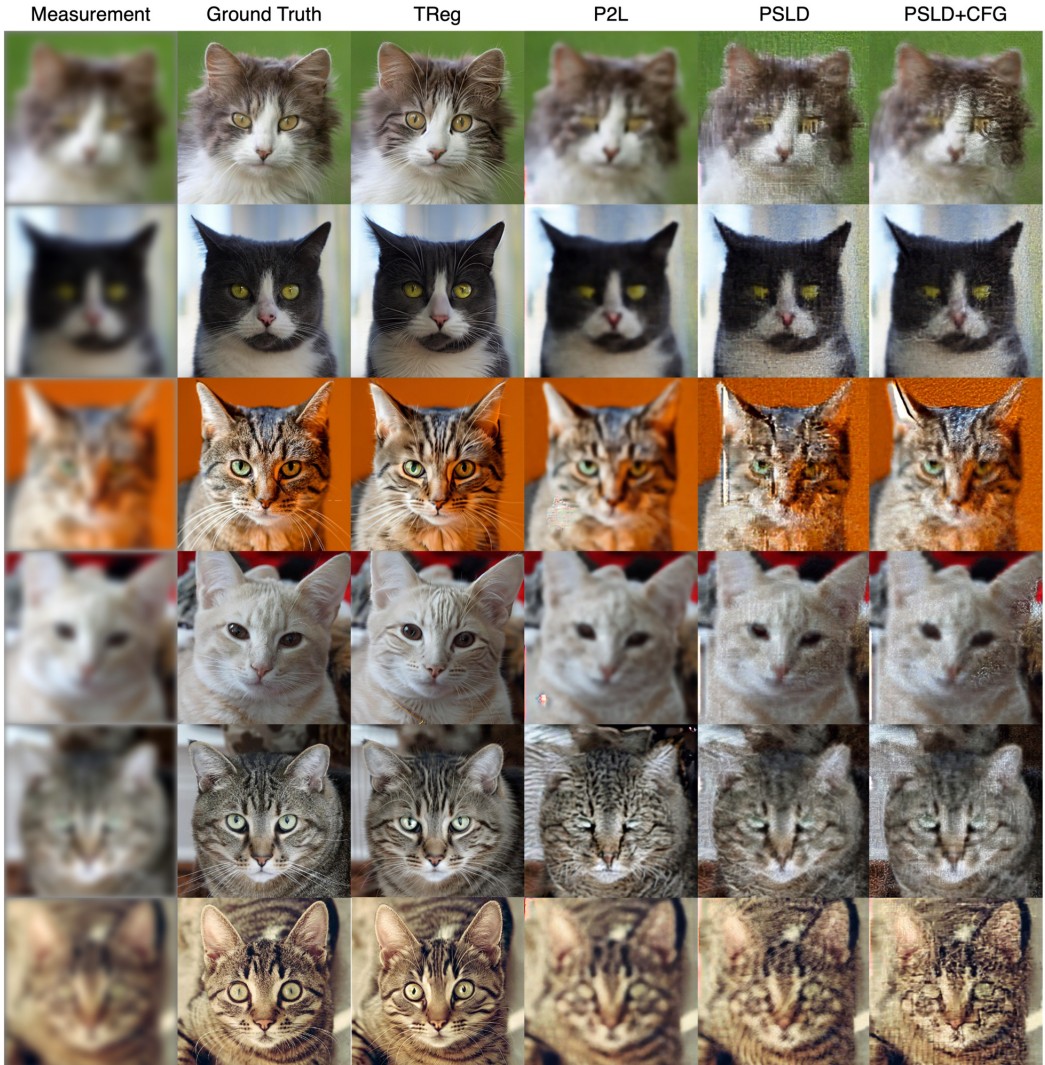

Figure 15: Qualitative comparison for deblurring (Gauss) task on AFHQ

## J    LIMITATION

While TReg demonstrates its performance in breaking the symmetry in the Fourier Phase Retrieval problem, its effectiveness is contingent upon the text prompt. For instance, when the true image features a complex or colorful background, a simple text prompt such as "a photography of a face" may not suffice to reconstruct a unique solution without residual symmetry.

In our scenario, we assume that the user can access the category of the dataset (e.g., FFHQ for faces). However, in real-world scenarios, it may be challenging to derive more informative text prompts solely from measurements with severe degradation. In general, finding suitable text prompts remains an open problem in text regularization for solving inverse problems.

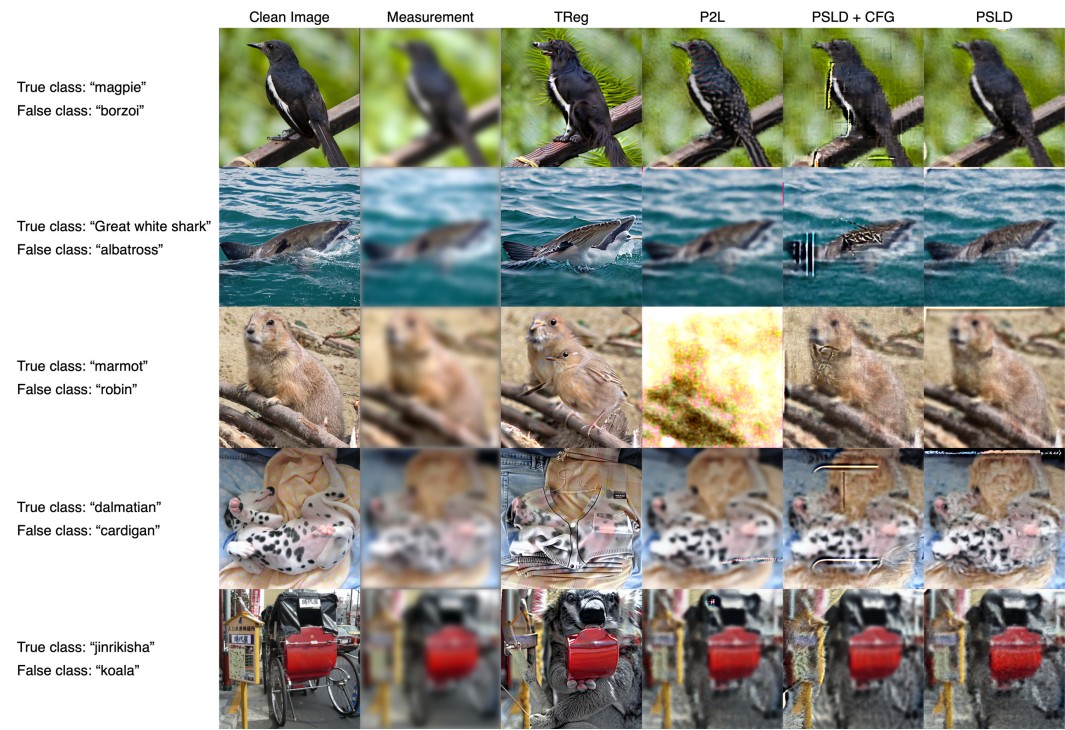

Figure 16: Reconstructions for deblurring task when given text prompt differs from the original class. ImageNet validation set (1k images) is leveraged.

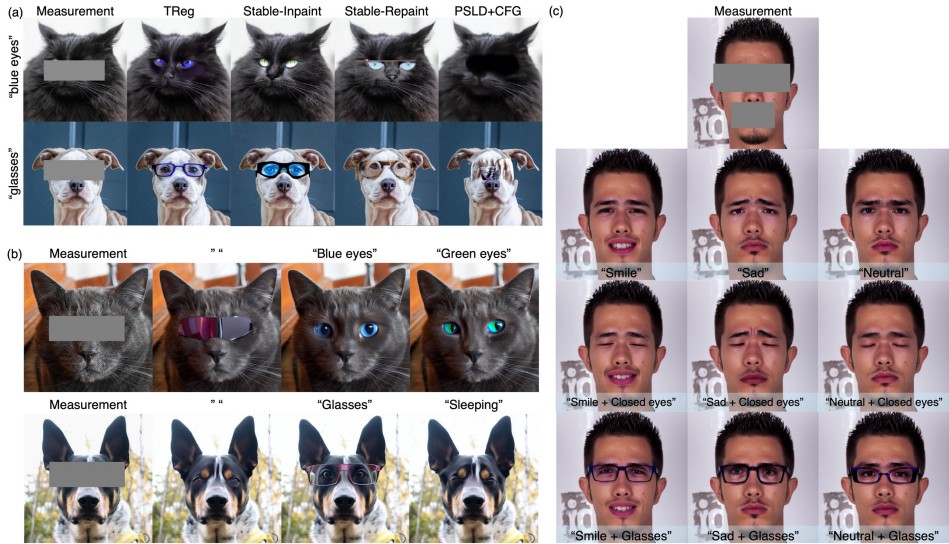

Figure 17: Representative results for box inpainting task. (a) TReg is better at finding solution according to text with high-fidelity. (b) Text regularization helps to reconstruct as intended. (c) TReg is possible to reconstruct based on multiple concepts.

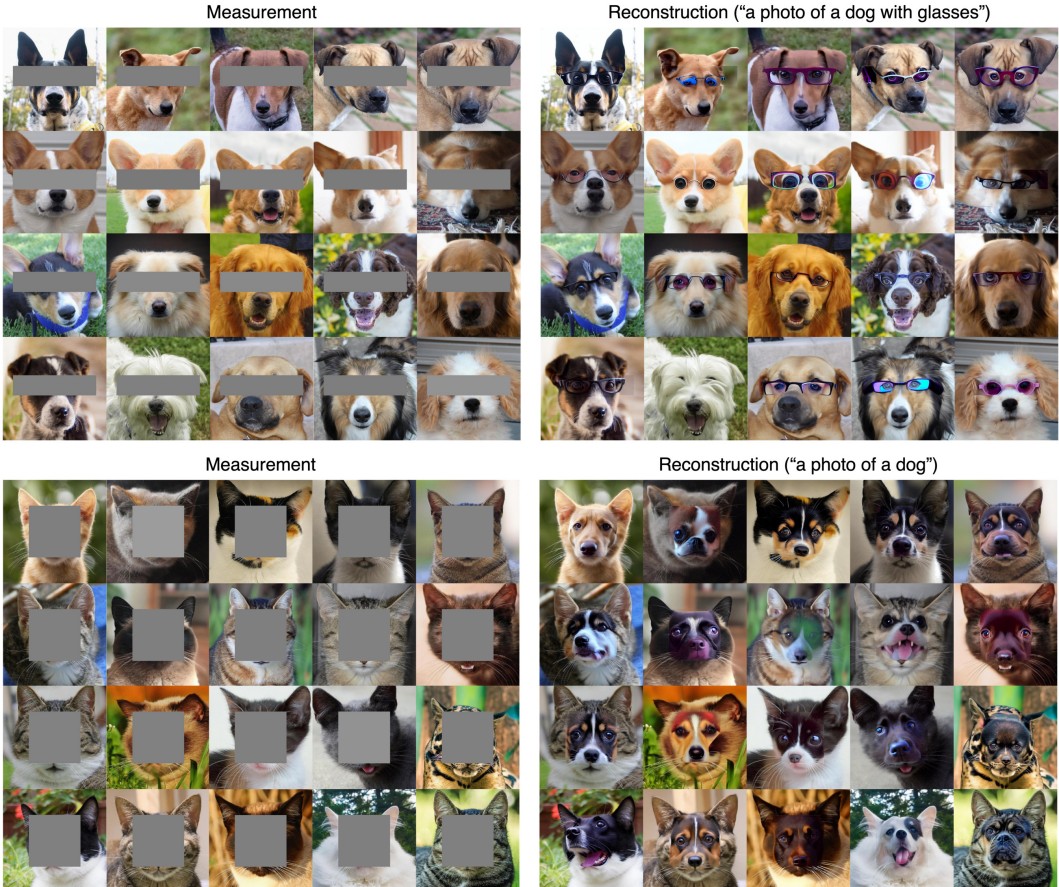

Figure 18: Uncurated results for box inpainting task. (Top) Inpainting of eye region of AFHQ-dog with the prompt "a photo of a dog with glasses". (Bottom) Inpainting of face region of AHFQ-cat with the prompt "a photo of a dog".

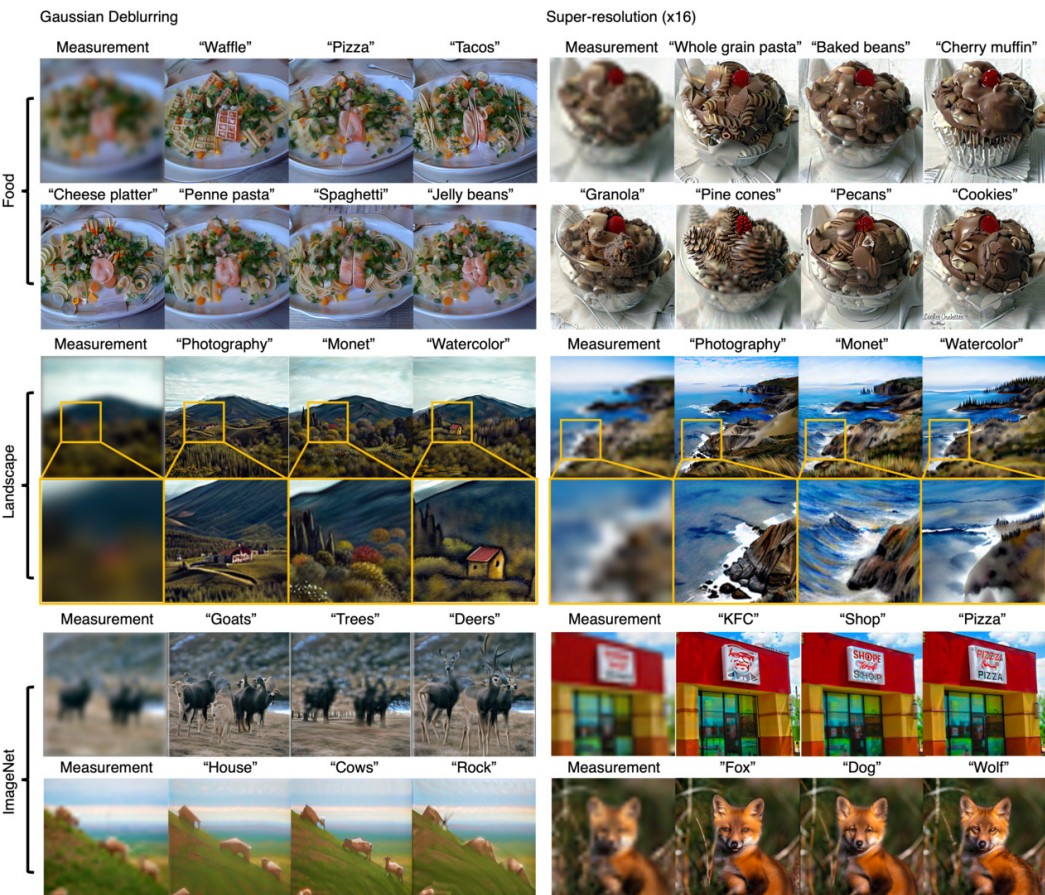

Figure 19: Reconstructions for diverse domain.

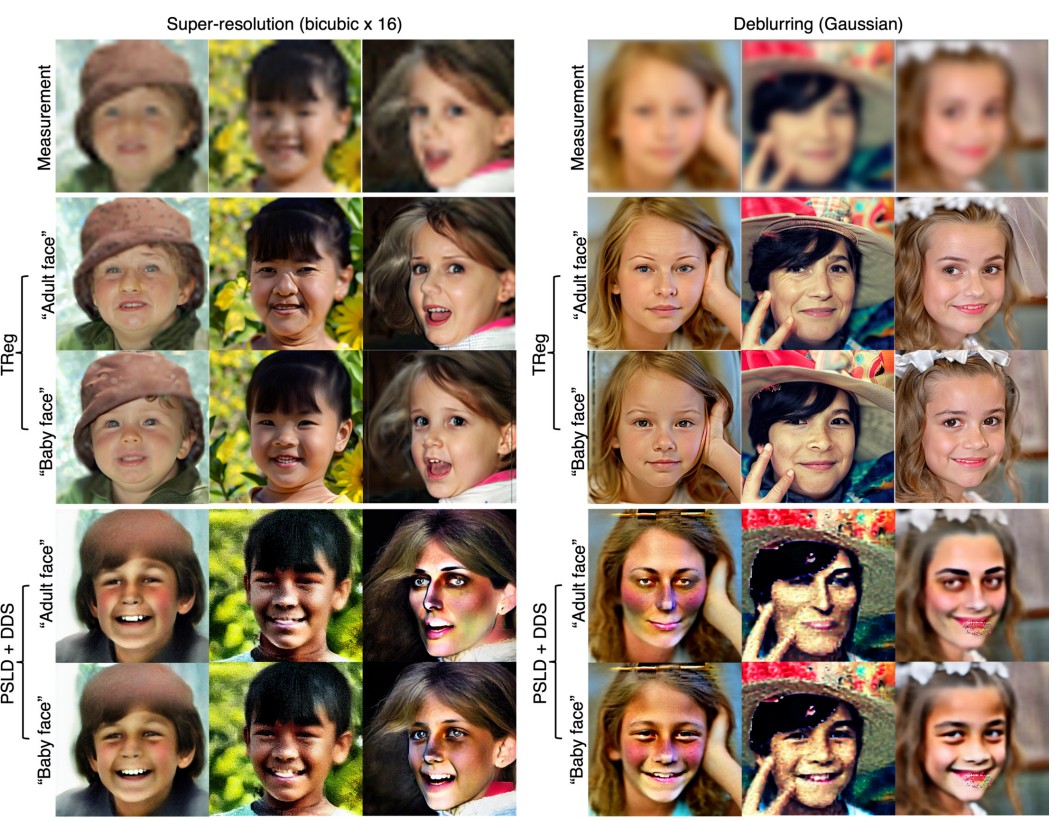

Figure 20: Reconstruction result for SR (x16) and Deblur (Gaussian) task on FFHQ dataset. Our method outperforms the existing methods (PSLD+DDS) in terms of the reconstruction quality.

