# OpenReview forum: "Regularization by Texts for Latent Diffusion Inverse Solvers"
_ICLR.cc/2025/Conference — ICLR 2025 Spotlight_

### Official Review · Reviewer_2SWh · 2024-11-01

**Soundness:** 3
**Presentation:** 2
**Contribution:** 3
**Rating:** 6
**Confidence:** 4

**Summary:**

This work uses diffusion models with text guidance in order to solve inverse problems. Inverse problems (e.g., in-painting or deblurring) have been recently tackled using diffusion models, however, their ill posedness nature makes resolving some ambiguities still challenging. The authors propose to leverage a user bias via a text prompt in order to guide the diffusion process towards a solution that is aligned with the description provided. The proposed framework uses a latent diffusion model with classifier-free guidance as a core framework with two innovations compared to previous works: (1) a term that induces alignment between the latent being optimized and that conditioned through the text prompt, and (2) a dynamic null-text adaptation. (1) is the distance between the latent embeddings that are being optimized and those obtained with the text used as guidance (in a classifier free guidance algorithm). (2) The null-text (which usually is the representation provided without any guidance) is updated dynamically in order to be aligned with the CLIP embeddings of the text provided by the user, incentivizing in this way the embeddings to continue to learn new concepts by moving the null-text closer and closer to the text description.

The authors show results on super resolution, deblurring and Fourier Phase Retrieval problems and compares with some baselines such as DDRM, PGDM, PSLD, and PnP.

**Strengths:**

The paper tackles an interesting problems with many potential applications.

While the individual ingredients are not completely new, even in the context of inverse problems (the use of prompts and the idea of solving the problem using alternate direction method have been explored by Chung et al in “PROMPT-TUNING LATENT DIFFUSION MODELS FOR INVERSE PROBLEMS”; the optimization of the null-text has been explored by Mokady et all in “Null-text Inversion for Editing Real Images using Guided Diffusion Models”) the overall framework and the way these ingredients are used is novel and seems to lead to compelling results.

**Weaknesses:**

The main weaknesses of this work are in the lack of quantitative evaluations on full datasets, lack of some important baselines and the clarity of the writing could be improved. See below for more detailed comments.

Evaluation: all tasks presented are obtained on specific classes rather than being averaged across full datasets as it is done in other the state of the art works (e.g., P2L). Specifically:
- Ambiguity reduction: it is unclear if the quantitative evaluation in Figure 3 is an average over a dataset or simply the results of the image shown (bear). It is also unclear from which source this image comes from. Maybe it is ok since this is an interesting experiment but not the core? In any case more clarity is needed.
- Accuracy of obtained solution: use true class as c. Results in  Table 1 and Table 2 show quantitative measurements (great) for two tasks (super resolution and de-blurring) but only on two classes “Ice-cream” and “fired rice” rather than averaging across full datasets.
- Accuracy of obtained solution: use different class as c. Results shown in Table 3 are for only two classes using two different C labels (fried rice to spaghetti, and Ice Cream to Macaroons). It is understandable to show selected classes in Figures but the quantitative evaluation should be reported on a larger datasets (with mean and standard deviations).
- Results on non-linear inverse problems. These results are only qualitative shown on 4 different images (Figure 6).

This work shares a lot of similarities with P2L (Chung et al.) so I was expecting P2L to be among the baselines. In the intro when talking about P2L the authors say “However, this method primarily focuses on data consistency and lacks robust alignment with textual prompts.” It would be great to support this statement with quantitative results.

For the alternate directions method there should be some additional information about convergence. Being an iterative method one would expect to see some analysis about it but I did not find any information.

The work of Mokady et al also shares the idea of adaptive negation (but with different application and implementation) so for Inpainint experiments (which are suitable for Mokady) this would also be a useful baseline.

Writing: The paper is not alway easy to follow. The readability of the manuscript would benefit from some re-writing. For example
- Background session could be summarized and moved to the appendix since these are all known concepts.
- The additional space gained should be used to provide more details about the proposed method. Here are some examples of things that could be clarified but there are more throughout the text:
    - Probably the most important: authors should mention that the proposed update is scheduled to be performed every so many iterations  end not always. This is somewhat hidden in the “range” argument of algorithm 1 but it becomes clear only reading the appendix. I strongly encourage the authors to bring this discussion into the main paper.
    - Consider moving the inpaiting results (with comparison with Mokady) in the main paper.
    - In Eq (13) and (14) there is the term A(x). Given that A is an “imaging system” but it is not clear the role it plays in practice when solving a specific problem. I would recommend the authors to provide some information about the role of the forward operator when discussing section 3.1.
    - “Here, ζ,γ are empirically chosen to satisfy¯αt−1 = ζ/(ζ+ γ).” Please provide an explanation why satisfying this equation is important.
    - …
- Since Figure 1 is even before the abstract the reader should be able to understand it without reading anything but the caption and looking at the figure itself. Consider increasing the details in the caption or moving the figure in a place where the reader has sufficient information to understand all of the examples.
- I find the abstract lacking important details. It should encourage a reader to continue the reading but in order to do so the more details  need to be presented. Here some potential improvements:
    - ``ambiguities in measurements or intrinsic system symmetries.’’ consider adding a concrete example of these problems.
    - define what adaptive negation is  or avoid adding a non-defined term.
    - ``Our comprehensive experimental results’’ consider telling the reader which are the most important quantitive results so that it encourages the reader to continue.
- In the experimental section consider breaking down the baselines for each experimental task. Currently they are presented all at once at the beginning irrespective of the tasks and I find this to be too much information to process and too far from where it is useful.
- The authors used the prompt “A photography of a ….”. This might not be grammatically correct. Photography is the art of creating photos or the process of creating photos. A better prompt would  be “A photo of a…” or “A photograph of a …”. I would be curious to know if this changes the results.
- Tweedie’s formula should be cited.
- From (10) to (13) the terms are swapped. Please consider maintaining consistent order.
- While the fact that the work uses a pre-trained VAE is mentioned in the main paper, I think the fact that also the diffusion model is pre-trained is not explicitly mentioned before the appendix. Consider clarifying this aspect in the main paper.
- The following sentences are unclear, consider rephrasing them:
    - “We also prepare a situation that different class label is given as a text description. Here, we should carefully set proper text prompt for measurement to avoid ignoring the provided guidance”.
    - “For the case that given text description is differnet from the original classes, we use “spaghetti" and “macarons", respectively.”. There is a typo (differnet) but beyond that it is unclear what this sentence means.
    - “For data consistency methods, we utilize the measurement itself “. What is the measurement?
    - “TReg leads to a unique solution corresponding to the given text description”. Maybe “TReg leads to similar solutions.”? Possibly one could say “semantically unique and aligned with the text”?
    - “ This discrepancy is clearly observed in pixel-level uncertainty in Figure 3(c). “ I suppose this is the variance computed across the 10 reconstruction? Consider mentioning explicitly how uncertainty is computed.

**Questions:**

Main questions (answering these questions could affect the recommendation)

- Could the authors provide information about convergence of the alternate direction method?
- Could the authors provide quantitive evaluation averaged across full datasets (as done in state of art works, e.g., P2L) and include relevant baselines, as mentioned above P2L should be included. Mokady would also be a good comparison for inpainting results.

Other questions (this are more curiosity driven questions, it is unlucky the answers here will change the recommendation)
- In figure 2 authors show that thanks to the adaptive negation strategy the generated image is less noisy. It is unclear however why this would be the case. Could it be that without adaptive negation one could reach a similar results with more iterations?
- Due to (18) does it mean that the VAE needs to have the same dimensionality of CLIP embeddings?
- “The proposed solver without adaptive negation tends to achieve higher PSNR than with adaptive negation, since it obtains blurry images”. Could the Structural similarity index measure (SSIM) be a better metric in this case?
- For the phase retrieval experiments: the error in the LDPS reconstruction seem to go beyond the ambiguity of symmetries… Do the authors have any intuition about why the quality of LDPS reconstruction is so low compared to TReg?
- How sensitive is this frameworks to slight variation of the text? This would be an interesting discussion of the paper. The authors use “A photography of …” how much do results change if instead they used “A photo of xxx” or simply “xxx”?

---

> ### Author Response · Authors · 2024-11-21
> **Response by authors (1/4)**
>
> **W1. Ambiguity reduction: it is unclear if the quantitative evaluation in Figure 3 is an average over a dataset or simply the results of the image shown (bear). It is also unclear from which source this image comes from.**
>
> **A**. We would like to clarify that the variance plot in Figure 3 is drawn by repeating reconstruction 10 times with a single measurement, where the true image is sampled from the ImageNet validation set. We added this detail in the revised manuscript. As our goal was to provide a proof-of-concept demonstrating the benefits of text regularization in reducing ambiguity, we believe this level of detail is sufficient. In fact, the remaining part of the experiments demonstrates the ambiguity reduced by TReg. We added the detail in the revised manuscript.
>
> **W2 & Q2. Accuracy of obtained solution: use true class as c. Results in Table 1 and Table 2 show quantitative measurements (great) for two tasks (super resolution and de-blurring) but only on two classes “Ice-cream” and “fired rice” rather than averaging across full datasets.**
>
> **A**. Thank you for the valuable comment. In Table 1 of the general comment, we evaluated TReg on the full validation sets of FFHQ, AFHQ-cat, and AFHQ-dog—widely used benchmarks for inverse problem solvers—using the true class as a text condition per Reviewer’s request. These datasets provide representative and measurable results compared to full Food-101 validation, which includes over 100 classes. To strengthen the comparison, we also included P2L as a baseline (Please refer to W5 & Q2 and Appendix C for more discussions on P2L). Table 1 demonstrates that TReg consistently outperforms other baselines across all inverse problems with high ambiguity. This advance stems from the given text conditions which effectively narrow down the solution space. These results have been added to Table 5 in the revised manuscript.
>
>
> **W3. Accuracy of obtained solution: use different class as c. Results shown in Table 3 are for only two classes using two different C labels (fried rice to spaghetti, and Ice Cream to Macaroons). ...the quantitative evaluation should be reported on a larger datasets (with mean and standard deviations).**
>
> **A**. Thank you for the thoughtful and valuable feedback. As requested, we have compared TReg and the baselines on the full FFHQ and AFHQ validation sets, widely used benchmarks for inverse problems, with true class text conditioning (Table 1 in general response). Additionally, we conducted more experiments on Food-101 subsets (e.g., “risotto → french fries” and “gnocchi → spaghetti”), representing practical scenarios where perceptually estimated (non-ground-truth) classes are provided as text cues (Table 2 in general response). We believe these results are both representative and measurable, especially compared to the full Food-101 validation set of over 100 classes. Given that most diffusion-based inverse problem solvers typically use 2-4 datasets, we would like to gently assure the reviewer that our evaluation is sufficient to demonstrate the method's effectiveness. Please see the general comments and responses to W5 and Q2 for further discussion.
>
> We would like to gently clarify that our evaluation pipeline is carefully designed to assess both (a) data consistency and (b) textual alignment simultaneously. This differs from image editing tasks, which often prioritize preserving coarse level of spatial layout over strict alignment with measurements. In our context, reconstructed solutions must align with noisy measurements while remaining perceptually plausible for human users. Evaluating whether measurements can be “translated” into clean solutions for arbitrary classes is beyond the scope of our work and would require measurement-specific benchmarks. We are excited to see future works in this direction.
>
> **W4. Results on non-linear inverse problems. These results are only qualitative shown on 4 different images (Figure 6).**
>
> **A**. We would like to assure the reviewer that they are proof-of-the concept experiments to show the TReg also works for non-linear cases, which is different from  two main inverse problems in our paper.  In the final version of the paper, we will try our best to include quantitative results with more extensive experiments.

---

> ### Author Response · Authors · 2024-11-21
> **Response by authors (2/4)**
>
> **W5&Q2. This work shares a lot of similarities with P2L (Chung et al.) ... In the intro when talking about P2L the authors say “However, this method primarily focuses on data consistency and lacks robust alignment with textual prompts.” It would be great to support this statement with quantitative results.**
>
> **A**. We kindly remind the reviewer that P2L is designed to update text prompts to enhance data consistency. While P2L can initialize the text prompt, this functionality is solely aimed at achieving better consistency with the measurement and does not facilitate finding a solution that aligns with the given text prompt.
>
> Please refer to Appendix C of revised manuscrip for more discussions on P2L. As mentioned in lines 78-82, we highlight that P2L is orthogonal to our approach, despite their apparent similarities. Specifically, P2L merely treats the null-text prompt as an additional parameter to improve the data consistency, optimizing it as:
>
> $\mathcal{C}^* = \arg \min_{\mathcal{C}} \| \mathbf{y} - \mathcal{A}(\mathcal{D} (\hat{\mathbf{z}}_{0|t}(\mathcal{C}))) \|^2 $ (Eq. 12 in P2L).
>
> Unlike TReg, P2L does not guide outputs toward a specific mode aligned with semantic linguistic conditions. It lacks support for Classifier-Free Guidance (CFG) and relies solely on conditional scores. While both methods leverage prompt embeddings, they do so in fundamentally different ways: P2L focuses on data consistency, whereas TReg reduces the solution space by incorporating rich perceptual estimates of noisy measurements in the form of linguistic descriptions.
>
> To adapt P2L as a text-based baseline, we assume an ideal scenario where the ground-truth class is provided for the noisy measurement. Please refer to the general comments for the results on the full validation set of FFHQ and AFHQ. The text embedding is optimized by using the Adam optimizer (learning rate 1e-4) with K=1. We perform reverse sampling with 200 timesteps, consistent with other baselines. That said, we note that P2L requires twice the neural function evaluations (NFEs) of TReg, as it recomputes Tweedie estimates after updating the text prompt. Further runtime analysis is included in the Appendix.
> Moreover, to fully address reviewer concerns, we also evaluate P2L in a more practical scenario where the estimated (non-ground-truth) class is provided as text, simulating real-world user text cues. As shown in Table 3 of the response and Figure 5 in the revised manuscript , P2L and other baselines fail to align outputs with the provided text, as they do not fundamentally encourage alignment with the given text. Please refer to W6 & Q1 for more comparative analysis on the sampling process.
>
> [Table 3].  Food - main experiment, comparison with P2L  / NFE 200
>
> |  | IceCream → Macaron (SR) |  |  | IceCream → Macaron (Deblur) |  |  | FriedRice → Spaghetti (SR) |  | |  FriedRice → Spaghetti (Deblur) |  |  |
> |:-:|:-:|:-:|:-:|:-:|:-:|:-:|:-:|:-:|:-:|:-:|:-:|:-:|
> |           | y-MSE |  LPIPS  | CLIP sim. | y-MSE | LPIPS | CLIP sim. | y-MSE |  LPIPS  | CLIP sim. | y-MSE | LPIPS | CLIP sim. |
> | P2L       | 0.004 | 0.747 | 0.229 | 0.023 | 0.758 | 0.264 | 0.005 | 0.758 | 0.231 | 0.027 | 0.784 | 0.253 |
> | TReg      | 0.004 | 0.771 | 0.314 | 0.011 | 0.743 | 0.312 | 0.005 | 0.769 | 0.303 | 0.013 | 0.737 | 0.300 |

---

> > ### Comment · Reviewer_2SWh · 2024-11-21
> > **Re: Response by authors (2/4)**
> >
> > Thank you for providing these results. I agree that P2L is different than TReg but as your results show it is still a reasonable baseline. I believe comparing to P2L strengthen the results of your work.

---

> ### Author Response · Authors · 2024-11-21
> **Response by authors (3/4)**
>
> **W6 & Q1.  For the alternate directions method there should be some additional information about convergence. Being an iterative method one would expect to see some analysis about it but I did not find any information.**
>
> **A**. We thank the reviewer for the constructive comment. Our approach optimizes the full objective in an alternating manner, where the data consistency is optimized in pixel space and the text-conditional regularization is mainly applied in latent space. A direct theoretical convergence analysis is challenging due to the complexity arising from the transition between pixel and latent spaces. However, in the Appendix B of the revised manuscript, we provide an informative empirical analysis by evaluating the progression of two objectives during the reverse sampling process: data consistency in pixel space and the denoising score matching loss (i.e. score distillation sampling loss) conditioned on a given text prompt.
> Specifically, we plot $\| y - A(\hat x_{0|t}) \|^2$ for data consistency, which measures how well the solution aligns with the given measurement. Also, we measure $| \epsilon - \epsilon_\theta (x_t, t, c) |^2$, a text-conditional diffusion model objective which evaluates how well solutions align with the clean data manifold and the corresponding text condition c. We present results for two scenarios: when the text prompt matches the ground-truth label and when it differs. For both cases, we plot the mean and standard deviation of these objectives, averaged over 100 samples.
> As shown in Figure 8 in the appendix of the revised manuscript, both objectives decrease significantly during the reverse sampling process, demonstrating that the TReg empirically promotes convergence. Notably, the data consistency of TReg improves particularly in the later sampling phase, which is in line with the interpolative constant setting in eq (13) of the revised paper. Please refer to W9-4 response for more detail.
>
>
> **W7. The work of Mokady et al also shares the idea of adaptive negation (but with different application and implementation) so for Inpainint experiments (which are suitable for Mokady) this would also be a useful baseline.**
>
> **A**. NTI is fundamentally different from TReg although they use the null-text embedding as a parameter for updates. NTI focuses on editing tasks, where the null-text embedding is updated to ensure that the reverse sampling trajectory does not deviate significantly from the pivot trajectory generated by DDIM inversion with a conditioned score function (i.e. CFG guidance scale of 1.0). This null-text is designed to preserve the contents of the source image while incorporating the editing text prompt. Therefore, it does not involve suppressing complementary concepts, as achieved by adaptive negation. More importantly, TReg does not incorporate the inversion process and also NTI does not incorporate data consistency problems.
>
> **Writing**
> We really appreciate the reviewer for carefully reading our manuscript and providing comments to enhance the readability. We fixed it by considering feedback from the reviewer. Also, we would like to clarify some unclear phrases that the reviewer mentioned point-by-point in below.
>
>
> **W8. Background session could be summarized and moved to the appendix since these are all known concepts.**
>
> **A**. Thanks for the suggestion. We moved section 2.2 (classifier-free-guidance) to the appendix. However, section 2.1 and 2.3 are crucial as they provide the context of this study which is important for one who is not familiar with diffusion-based inverse problem solving, so we maintain them in the background.
>
> **W9. The additional space gained should be used to provide more details about the proposed method.**
>
> - authors should mention about update range in the main paper
>
>     We would like to gently clarify that it is common to update the latent code “every” iterations during the reverse process in diffusion-based inverse solvers, including DDRM, DPS, PiGDM, PSLD, and even for P2L. However, we just skip some steps for computational efficiency. That being said, we clarify the details of the update range in the Appendix because it is optional.
>
> - Consider moving the inpaiting results (with comparison with Mokady) in the main paper.
>
>     Due to page limitations, we provide a representative example in Fig. 1, and refer readers to the appendix for additional results per reviewer’s request. For the comparison with Mokady, please refer to our response to W7.
>
> - Term A(x) in Eq (13) and (14): it is not clear the role it plays in practice when solving a specific problem.
>
>     We would like to gently remind the reviewer that it is a conventional term in inverse problems. However, we have revised the phrase to "forward measurement operator" for clarity.

---

> > ### Comment · Reviewer_2SWh · 2024-11-21
> > **Re: Response by authors (3/4)**
> >
> > **W6 & Q1.** Thank you for providing the empirical converge analysis.
> >
> > **W7. The work of Mokady et al** I agree with you that the work in Mokaday et al. is not the same at TReg, however, it has some similarities and since of the the claims in the paper is that ``TReg is comparable or superior compared to state-of-the-art image editing algorithms,`` I think that Mokaday could be considered one such state-of-the-art image editing algorithms. If a comparison is not feasible due to time I would consider toning down the claim above.
> >
> > **W9. Range** You mentioned in the answer that you skip some steps for computational efficiency but from the appendix it seems to imply more than just computational efficiency: ``Our empirical observation show that this partial data consistency update achieves better trade-off between the image reconstruction consistency and the latent stability.`` Could you please clarify is performing this every 3 steps (or 10 for the phase retrieval) is just a computational efficiency trade-off or if there is some other implications rated to consistency? How robust is the algorithm to the choice of this parameter?
> >
> > Minor: You call this hype parameter of your algorithm *Range*. To me range implies that you perform something continuously for the *range* from time A to time B, while this parameter is something that controls how often you perform the update rather than for how long. Maybe *Frequency* could be more intuitive name for this parameter.

---

> ### Author Response · Authors · 2024-11-21
> **Response by authors (4/4)**
>
> **W9. (continue)**
> - “Here, ζ,γ are empirically chosen to satisfy¯αt−1 = ζ/(ζ+ γ).” Please provide an explanation why satisfying this equation is important.
>
>     $\zeta$ and $\gamma$ are hyperparameters for the proximal optimization problem, chosen to satisfy the second equality in (15). Specifically, we adjust the interpolation coefficient to prioritize $\hat{z}_0(y)$ during the final phase of reverse sampling, ensuring fine-grained refinement for improved data consistency. Empirically, this strategy demonstrates robust performance, as described in the revised manuscript. This is potentially attributed to the coarse-to-fine detail refinement inherent to reverse diffusion sampling.
>
> - In the experimental section consider breaking down the baselines for each experimental task. Currently they are presented all at once at the beginning irrespective of the tasks and I find this to be too much information to process and too far from where it is useful.
>
>     About baselines, we wrote their features in `lines 349-361`.
>
> - The authors used the prompt “A photography of a ….”. This might not be grammatically correct. Photography is the art of creating photos or the process of creating photos. A better prompt would be “A photo of a…” or “A photograph of a …”. I would be curious to know if this changes the results.
>
>     Negligible. Please see the response to Q7.
>
> - Details in Figure 1 caption. Tweedie’s formula should be cited. From (10) to (13) the terms are swapped. Consider clarifying the diffusion model is pre-trained. Rephrase suggestions.
>
>     Fixed.
>
>
>
> **Q3**. In figure 2 authors show that thanks to the adaptive negation strategy the generated image is less noisy. It is unclear however why this would be the case. Could it be that without adaptive negation one could reach a similar results with more iterations?
>
> **A**. Because the prior given by text description corresponds to clean images, adaptive negation is also effective to reduce blurry effects which do not exist in clean images. Even though it is possible to reach similar results with more iterations instead of adaptive negation, it is beneficial to use the adaptive negation as it may accelerate problem solving.
>
> **Q4**. Due to (18) does it mean that the VAE needs to have the same dimensionality of CLIP embeddings?
>
> **A**. We guess that the reviewer is pointing out the equation (19). As noted in `line 282`, $\mathcal{T}_{img}$ denotes the CLIP image encoder, which maps the image to CLIP embedding space. Therefore, VAE does not need to have the same dimensionality of CLIP embeddings.
>
> **Q5**. “The proposed solver without adaptive negation tends to achieve higher PSNR than with adaptive negation, since it obtains blurry images”. Could the Structural similarity index measure (SSIM) be a better metric in this case?
>
> **A** We observed that SSIM shows the same tendency as the PSNR. Thus, we reported FID which indicates the distance of two distributions.
>
> **Q6**. For the phase retrieval experiments: the error in the LDPS reconstruction seem to go beyond the ambiguity of symmetries… Do the authors have any intuition about why the quality of LDPS reconstruction is so low compared to TReg?
>
> **A**. Because we are conducting all experiments with 200 NFEs, while both DPS and PSLD have used 1000 NFEs in their paper. Reducing the number of timesteps can cause insufficient update of latent code and results in lower quality of reconstructions. In contrast, TReg can effectively solve the problem within 200 NFEs which is beneficial in its efficiency.
>
> **Q7**. How sensitive is this frameworks to slight variation of the text? This would be an interesting discussion of the paper. The authors use “A photography of …” how much do results change if instead they used “A photo of xxx” or simply “xxx”?
>
> **A** We appreciate the reviewer’s constructive feedback. While the overall performance is not highly sensitive to grammatical details, we observed marginal improvements when adjusting the prompt as kindly suggested by the reviewer (Table 4). This observation suggests that TReg can potentially benefit from carefully crafted prompts.
>
>
> [Table 4] Sensitivity to text prompt. Evaluated on Deblurring task with AFHQ-cat.
>
> |   |  FID  | LPIPS |
> |:--:|:--:|:--:|
> "a photo of a cat" | 47.70 | 0.348 |
> "a photography of a cat" | 48.08 | 0.347 |
> "a cat" | 50.53 | 0.353 |

---

> > ### Comment · Reviewer_2SWh · 2024-11-21
> > **Re: Response by authors (4/4)**
> >
> > Thank you for all the responses.
> >
> > One minor suggestion:
> > Since you mentioned that it is possible to reach similar results with more iterations instead of adaptive negation, it would be great if you could quantify the benefit. How many more iterations would be needed? If you could show that you reach the same results with a small fraction of interactions that would strengthen your results.

---

> ### Comment · Reviewer_2SWh · 2024-11-21
> **Re: Response by authors (1/4)**
>
> Thank you for your reply and for considering my suggestions. I would like to start by highlighting that despite the initial rating I like this work. My comments are only aiming to improving it further to the point that it shows stronger scientific rigor. I hope my feedback can help towards making the paper stronger, I am not trying to discredit this work.
>
> **Re W2 & Q2**
> Thanks for providing more quantitative results and a comparison with the highly related P2L baseline. While I was expecting something like P2L Table 2 and 3 (i.e., FFHQ and ImageNet on super resolution, motion deblur,  gauss deblur, and inpainting). That said,  the new results can strengthened a bit the original submission. The weakness of the new results is that they represent only 3 classes (persons, cats and dogs) and it is unfortunate they do not include inpainting. Following the experimental settings of P2L would have shown results on the whole variability of ImageNet which would have been even more convincing.
>
> **Re W3**
> Thank you for adding a few more classes to your experiments. Similarly to my sentiment above: I appreciate the effort you placed into performing more experiments but this is not exactly what I was asking for. I was asking for quantitative evaluation on full datasets. In my opinion to support the claims of this section one would need to show results on the full dataset translated to at least 1 other class. Without this a reader will always be left asking if these were hand picked examples or if the algorithm really generalizes.
>
> Statements like ``TReg is comparable or superior compared to state-of-the-art image editing algorithms, highlighting the efficacy of text-guidance facilitated by TReg``, or ``TReg effectively integrates the given text prompt
> while preserving robust data consistency, thereby validating its efficacy``, would carry a stronger message if they were supported by strong evidence rather than evaluating the algorithm on a handful of classes.
>
> Minor: Since you use the clip similarity score I would consider adding for every class you the average clip similarity of the original class (from validation set),  just to have a reference point (sine I assume even that similarity will not be 1.0).
>
> **Re W1 and W4**
> I appreciate that some experiments can be meant simply as a proof of concepts, however, I hope you can also appreciate that showing proof of concepts, while encouraging, is not sufficient to make any definitive claim.

---

> ### Comment · Reviewer_2SWh · 2024-11-21
> **Reviewed recommendation**
>
> As I mentioned in my answers above I appreciate the effort in providing more results and clarifications. Many aspects have been indeed clarified and the experiments added do provide a bit more evidence, although in my opinion, such evidence should be stronger. I do like the work but the experimental evaluation could and should be stronger. In light of this I am going to updated my scores and overall recommendation.

---

> > ### Author Response · Authors · 2024-11-24
> > **Additional Response to Reviewer 2SWh**
> >
> > We sincerely thank Reviewer 2SWh for their efforts and for carefully reviewing our responses and revisions. Below, we address the reviewer’s remaining concerns point by point.
> >
> > **Re W2&Q2.** As requested, we conducted an additional comparison for a box-inpainting task, where the mask is placed at the center of the image, covering half its size. We used FFHQ and AFHQ as our datasets and included PSLD, PSLD+CFG, and P2L as baselines. For the text prompt, we used the same descriptions as outlined in the general comments. As shown in Table 5, TReg outperforms the baselines in addressing box-inpainting tasks, which are particularly challenging for baseline models to achieve high-quality reconstructions. We added this result in Table 5 in Appendix G.
> >
> > [Table 5] True class as text prompt / Box Inpainting / NFE=200
> > | | FFHQ | | AFHQ | |
> > |:--:|:--:|:--:|:--:|:--:|
> > | | PSNR | FID | PSNR | FID |
> > |PSLD | $\underline{19.76}$ | **60.97**|16.93|104.7|
> > |PSLD+CFG|16.82|90.72|15.17|130.3|
> > |P2L|16.84 | 85.32 | 16.07 | 138.4 |
> > |TReg | **19.95** | $\underline{66.93}$ | **17.39** | **51.97**|
> >
> > **Re W3**
> > We agree that providing more robust experiments for text regularization would strengthen the manuscript. In response, we conducted an additional experiment solving SRx16 and Gaussian deblurring problems on ImageNet, using different classes as text descriptions.
> >
> > As noted in the general response, the experiments should assess whether the framework can reliably reconstruct “perceptually plausible” solutions from a single noisy measurement. To this end,  we construct a validation set from ImageNet comprising image-text pairs where the text description differs from the original class.
> >
> > First, we constructed a 1k ImageNet validation set encompassing all ImageNet classes, following the approach used in P2L. For each validation sample, the ground-truth class is known. Next, to identify a “perceptually plausible” text description, we measured pairwise LPIPS for all noisy measurements simulated from the 1k ImageNet validation set and selected the target text description corresponding to the class label with the lowest LPIPS samples. This process resulted in a 1k ImageNet validation set with both original and target classes (e.g. “great white shark” and  “albatross”, see Figure 16 in Appendix H). Finally, we solved the inverse problem on this validation set using the experimental setup outlined in the main experiment.
> >
> > Table 6 demonstrates that TReg effectively reconstructs solutions aligned with the given target class while satisfying data consistency, as evidenced by improved FID, CLIP similarity scores, and yMSE. In contrast, baseline methods exhibit significant trade-offs among these metrics. For example, PSLD achieves the best FID in the deblurring task, but its reconstructions fail to align with the given text description, as indicated by low CLIP similarity. Similarly, PSLD + CFG in the SR task achieves higher CLIP similarity but fails to maintain data consistency. In comparison, TReg delivers better or comparable performance across all metrics. These results highlight the versatility of text regularization with TReg in solving inverse problems for a wide range of natural images, extending beyond categories like person, cat, and dog. We also added this result in the Appendix H.
> >
> > [Table 6] Different class as Text prompt / ImageNet / NFE=200
> >
> > | |SR | | |Deblur | | |
> > |:--:|:--:|:--:|:--:|:--:|:--:|:--:|
> > | | yMSE | FID | CLIP sim | yMSE | FID | CLIP sim |
> > |PSLD | 0.010 | 168.6 | 0.219 | 0.025 | **113.0** | 0.215 |
> > |PSLD+CFG|0.024|116.5|**0.267**|0.025|148.8|$\underline{0.230}$|
> > |P2L|$\underline{0.006}$ | $\underline{143.5}$ | 0.213 | 0.030 | 118.8 | 0.215 |
> > |TReg | **0.002** | **111.1** | $\underline{0.259}$ | **0.025** | $\underline{113.9}$ | **0.267** |
> >
> > **Re W7**
> >
> > The statement ‘TReg is comparable or superior to state-of-the-art image editing algorithms’ was intended to emphasize the challenges of text-guided inverse problem solving, highlighting that a straightforward application of image editing algorithms is not feasible. In the revised manuscript, we have toned down this statement to more accurately reflect our intended meaning as requested.
> >
> > **Re W9**
> >
> > The update range is chosen for computational efficiency. The phrase ‘empirically working’ indicates that setting $\Gamma$ alongside other hyperparameters, such as the learning rate and the number of optimization iterations for the null-text, achieves sufficient performance. However, if $\Gamma$ is modified, the other hyperparameters may also need to be adjusted accordingly.
> >
> > **Minor suggestion on ablation study**
> >
> > Thanks for the suggestion. Given the time limit, in the final version, we will consider including an ablation study on the number of NFEs of diffusion models that can achieve comparable performance without adaptive negation.

---

> > > ### Comment · Reviewer_2SWh · 2024-11-25
> > > **Re: Additional Response to Reviewer 2SWh**
> > >
> > > Thank you for the additional experiments on ImageNet. Could you please clarify why you did not follow the setting in P2L and instead you used a text description that differs from the original class? What I was expecting is something that shows that the results presented in Figure 4 can generalize for many classes.

---

> ### Author Response · Authors · 2024-11-25
> **Re: Re: Additional Response to Reviewer 2SWh**
>
> Thanks for the prompt response. However, the reviewer’s comment is a bit confusing as in the previous round the reviewer said “ In my opinion to support the claims of this section one would need to show results on the full dataset translated to at least 1 other class”.
>
> If the reviewer is asking why we did not conduct an experiment of solving the inverse problem using true class as text description for ImageNet, we would like to assure the reviewer that  **we will ensure to include the results for ImageNet in the final version of the paper**. Due to the time constraints of the rebuttal period, we were unable to include comparisons with all baselines for all  tasks. But we tried our best to conduct additional experiments on box inpainting problems to address the reviewer's comments.
>
> On the other hand, if the reviewer is asking the reason for using text description that differs from the original class, the reviewer is kindly reminded that the primary focus of this paper is not solving conventional inverse problems. Instead, our aim is to demonstrate the feasibility of reconstructing solutions using text-based regularization. Consequently, we prioritized and spent our time validating the core contributions of our paper, as reflected in Table 6 of the main response and Figure 16 in Appendix H.
>
> Hope this addresses your question and please feel free to contact us if there are more things to clarify.

---

> > ### Comment · Reviewer_2SWh · 2024-11-25
> > **Re: Re: Additional Response to Reviewer 2SWh**
> >
> > Apologies for the confusion. I confused the results I was expecting for W2/Q2 (*"I was expecting something like P2L Table 2 and 3 (i.e., FFHQ and ImageNet on super resolution, motion deblur, gauss deblur, and inpainting)."*) with W3. I understand now that those results tackle W3 and I agree that they show more evidence for this use case.

---

> > > ### Author Response · Authors · 2024-11-27
> > > **Additional response to Reviewer 2SWh**
> > >
> > > Dear Reviewer 2SWh,
> > >
> > > Thank you for your timely response. We are pleased to hear that our additional experiment on W3 has addressed your concern.
> > >
> > > As the rebuttal period is extended, we conducted additional experiments on the ImageNet 1k validation set where the true classes are provided as text descriptions to address the remaining concern of the reviewer in W2&Q2.
> > > Specifically, we compared TReg, PSLD and P2L on Super-resolution (x16), Gaussian Deblurring, and Box inpainting tasks. In Table 7 below, TReg demonstrates superior performance compared to the other baselines. We added this result in Table 6 in the Appendix G. We also have updated Figure 4 in the revised manuscript to include examples from ImageNet, replacing the previous examples from AFHQ, as requested by the reviewer.
> > >
> > > Best regards, Authors
> > >
> > >
> > > [Table 7] Additional experiment on Imagenet, where true classes are given as text description
> > >
> > > |           | SR (x16) | | Deblur | |  Inpainting  | |
> > > |:-:|:-:|:-:|:-:|:-:|:-:|:-:|
> > > |           | PSNR | FID | PSNR | FID | PSNR | FID |
> > > | PSLD      | 18.01 | 170.5 | **20.97** | $\underline{115.9}$ | $\underline{17.41}$ |$\underline{90.13}$ |
> > > | P2L       | $\underline{18.62}$ | $\underline{141.1}$ | 19.58 | 117.0 | 15.94 | 119.3 |
> > > | TReg      | **19.71** | **69.65** | $\underline{20.66}$ | **55.92** | **18.11** | **50.67** |

---

> > > > ### Comment · Reviewer_2SWh · 2024-11-28
> > > >
> > > > Thank you for continuing to improve your work. All the new results provided during the rebuttal have alleviated my main concern regarding the evaluation protocol. I am happy to further upgrade my recommendation.

---

> > > > > ### Author Response · Authors · 2024-11-29
> > > > > **Response to the reviewer 2SWh**
> > > > >
> > > > > We are pleased to hear that our additional experiments have addressed the reviewer’s concerns. Thank you for your confirmation and for upgrading your recommendation.

---

### Official Review · Reviewer_vYBr · 2024-11-04

**Soundness:** 3
**Presentation:** 3
**Contribution:** 3
**Rating:** 8
**Confidence:** 3

**Summary:**

This paper introduces TReg (Regularization by Text), a method for solving inverse problems in image processing using latent diffusion models by conditioning on text.  TReg leverages user provided textual descriptions to guide the image reconstruction process.

Prior diffusion-based inverse solvers often struggle with ambiguity, as multiple different images can produce the same degraded output. TReg addresses this by incorporating textual prompts as a form of regularization. The authors frame the problem as a text-conditioned latent optimization, where the goal is to find a solution that is both consistent with the measurements and aligned with the provided text description.

The authors introduce the following:
* A text based regularization term that encourages the reconstructed image's latent representation to be close to a text-conditioned denoised estimate. This helps guide the sampling trajectory towards solutions that are semantically aligned with the text.
* To further refine the textual alignment, this method uses a novel "null-text optimization" technique. This dynamically adjusts the influence of the textual guidance during the reverse diffusion sampling process, suppressing unintended signal components.


The authors also discuss limitations, particularly the challenge of finding suitable text prompts for some challenging inverse problems where the problem is extremely ill posed.

**Strengths:**

Overall the paper is well written, provides clear motivation, and the authors perform reasonably comprehensive experiments. The authors do a good job in their logical flow and provide pretty clear explanations of the proposed method and experimental setup. Figures look nice.

In terms of novelty, they authors introduce a reasonably novel approach to solving inverse problems by incorporating text-based regularization into latent diffusion models.  This addresses a significant limitation of existing solvers, mostly their struggle with ambiguity, and mitigates the need for task specific training.

The proposed method is technically sound, with a clear mathematical formulation and well-defined optimization procedures.  The integration of LDPS further enhances the method's capabilities.

**Weaknesses:**

* `Line 071` In my view the connection to the human brain is quite weak, not sure why the authors put this in the paper...
* I would like to see additional, none cherry picked outputs. In my experience using diffusion models to recover images can often fail in very strange ways. I am specifically interested in image inpainting results, which nearly always result in boundary artifacts.
* I would like to see experiments in the presence of measurement noise, for example with JPEG compression artifacts after gaussian blurring.

**Questions:**

See above

---

> ### Author Response · Authors · 2024-11-21
> **Response by authors**
>
> **W1. `Line 071`: In my view the connection to the human brain is quite weak**
>
> **A**. We agree that `Line 71` does not provide core insight for the proposed method, so we excluded it in the revised manuscript. Thanks for carefully reading the manuscript.
>
> **W2. I would like to see additional, none cherry picked outputs. ...specifically interested in image inpainting results, which nearly always result in boundary artifacts.**
>
> **A**. We added uncurated samples for the inpainting problem in the Figure 17 of the Appendix of revised manuscript. Although there are few cases with remaining boundary artifacts for large box inpainting (Figure 17 bottom), TReg successfully mitigated boundary artifact issue.
>
> In case of inpainting, boundary artifacts occur due to inconsistency between in-mask and out-mask regions. To address this effectively, information of in- and out-mask regions should be mixed via algorithms like DPS [1, 2]. As noted in the section 3.2, TReg also requires steps of LDPS to elaborate this consistency. Nevertheless, TReg provides controllability on how to fill-out the masked region according to text description. On the other hand, a naive combination of PSLD and CFG did not work as shown in Figure 16 in revised Appendix.
>
>
> [1] Diffusion Posterior Sampling for General Noisy Inverse Problems
>
> [2] Solving Linear Inverse Problems Provably via Posterior Sampling with Latent Diffusion Models
>
>
> **W3. I would like to see experiments in the presence of measurement noise, for example with JPEG compression artifacts after gaussian blurring.**
>
> **A**. All the experiments are done with the presence of Gaussian noise to the measurement (`line 304`). Although the TReg shows the robustness to the measurement noise, we additionally tested another type of measurement noise: JPEG compression artifact. Specifically, we use the forward operator implemented by DDRM.  For JPEG compression, we vary the quality factor (QF) at 50, 30, and 10, where a higher quality factor indicates less compression.
>
> As shown in Figure 11 in Appendix, TReg demonstrates robustness up to Q30, providing nearly identical reconstructions. At Q10, however, significant noise is introduced into the measurement, resulting in degraded reconstructions. For such extreme JPEG artifacts, incorporating JPEG compression into the forward operation could be considered.

---

> > ### Comment · Reviewer_vYBr · 2024-11-21
> >
> > I thank the authors for adding the experiments regarding measurement noise in the revision, they are indeed helpful.
> >
> > I maintain my current score.

---

### Official Review · Reviewer_CtoC · 2024-11-08

**Soundness:** 3
**Presentation:** 3
**Contribution:** 2
**Rating:** 8
**Confidence:** 3

**Summary:**

This paper introduces a novel method for solving inverse problems using diffusion models, with a focus on reducing ambiguity in the solutions.
The authors propose a latent diffusion inverse solver that incorporates regularization by texts (TReg), which applies textual descriptions of the expected solution during the reverse sampling phase.
During training, this text guidance is dynamically adjusted through null-text optimization for adaptive negation.

Overall, I think this is a good paper.

**Strengths:**

* this paper introduces an explicit regularization term during training via text-driven regularization (TReg)
* the proposed TReg effectively addresses the challenge of ambiguity in inverse problems
* this paper further introduces an adaptive negation to dynamically adjust the influence of textual guidance
* the paper is well written and easy to follow, extensive in main text and supplementary demonstrate the effectiveness of the proposed method for diffusion inverse solvers

**Weaknesses:**

1. lack of overall results over whole dataset. all the experiments results are shown in visualizations or subset results of specific classes.
2. i notice in both table 1 and table 2, PSNR scores are higher without adaptive negation, do authors have any analysis or intuition about this results?
3. (this might not be a weakness, just naturally curious) in the experiments, the text regularization is only tested with class name, how about the results using some natural captions (such as generated with BLIP or GPT)?

**Questions:**

please refer to weakness part.

---

> ### Author Response · Authors · 2024-11-21
> **Response by authors**
>
> **W1. Lack of overall results over whole dataset**
>
> **A**. Thanks for the constructive comment. As requested, we compare TReg and the baselines on the full FFHQ and AFHQ validation sets–widely used benchmarks for inverse problems–with true class text conditioning in Table 1 of the general comment. To further simulate the unknown class scenario (where a perceptually plausible class is inferred and provided as text conditioning), we included additional experiments with more class subsets from the Food-101 dataset  (“risotto -> french fries“ and “gnocchi -> spaghetti“ ). In both cases, TReg demonstrates its superior performance in terms of image fidelity and textual alignment. For detailed results, please see the general comment.
>
> **W2. In both table 1 and table 2, PSNR scores are higher without adaptive negation, do authors have any analysis or intuition about this results?**
>
> **A**. As mentioned in the main paper (line 415-416) , PSNR tends to be higher when the image is blurrier, regardless of image fidelity. This is evident in Figure 2(b). Therefore, we reported a perceptual metric, FID, in addition to the PSNR.
>
> **W3. (this might not be a weakness, just naturally curious) in the experiments, the text regularization is only tested with class name, how about the results using some natural captions (such as generated with BLIP or GPT)?**
>
> **A**. Thank you for the insightful question. We believe natural captions generated by LLMs are compatible with TReg, offering the potential to enrich text conditions and automate the reconstruction process with reduced human intervention. That said, it is more crucial to investigate how to improve VLMs’ accuracy in inferring classes from noisy measurements. Moreover, Stable-Diffusion v1.5 involves CLIP as text encoder which limits the maximum context length and its compatibility with other advanced language models. We believe extending TReg with more powerful text encoders would be an interesting future work.

---

> > ### Comment · Reviewer_CtoC · 2024-11-23
> >
> > I thank the authors for adding the experiments and clarifying my concerns.

---

> > > ### Author Response · Authors · 2024-11-23
> > > **Thanks!**
> > >
> > > We are pleased to hear that our added experiments clarifies the reviewer's concern. Thanks for your confirmation.

---

### Author Response · Authors · 2024-11-21
**General Comment by authors**

We sincerely thank reviewers for their constructive, positive and thorough review. We are appreciate that the reviewer finds that our submission is **well-written** (`CtoC`, `vYBr`), effectively **addressing the interesting challenge of inverse problem** (`CtoC`, `vYBr`, `2SWh`) with **novel approach** (`vYBr`, `2SWh`), **technically sound** (`vYBr`) and **demonstrating its effectiveness** (`CtoC`). For point-by-point responses on the weakness and questions, please refer to our response below.

Here, we address the shared concern on the “experiments over the entire dataset” arised by reviewers `CtoC` and `2SWh`.

For the case where the true class labels are available as text, per reviewers’ request, we evaluate TReg on FFHQ, AFHQ-cat and AFHQ-dog validation set, since they are representative benchmarks on inverse problem solvers. For both cases, TReg outperforms all the baselines and demonstrates its effectiveness in solving inverse problems with high ambiguity (Table 1).

For the case where the different class is given as a text, which is a core experiment of the paper, we conducted additional experiments on Food-101 subsets (e.g., “risotto → french fries” and “gnocchi → spaghetti”) (Table 2). Here, TReg shows superior performance on additional evaluations, while baselines such as PSLD+PnP suffer from low data consistency exemplified by y-MSE.

We note that these experiments should evaluate whether the framework can reliably reconstruct “perceptually plausible” solutions from a single noisy measurement. Evaluating its ability to reconstruct solutions for “arbitrary” or “unrelated” categories lies beyond the scope of this work. Thus, we focus on representative subset pairs of visually related classes rather than the entire Food-101 dataset, which includes over 100 categories.



[Table 1]. FFHQ metric (“a photo of a person”) & AFHQ - ("a photo of a dog (cat)") / NFE 200

|           | FFHQ-SR | |FFHQ-Deblur | |  AFHQ-SR | |  AFHQ-Deblur |  |
|:-:|:-:|:-:|:-:|:-:|:-:|:-:|:-:|:-:|
|           | PSNR | FID | PSNR | FID | PSNR | FID | PSNR | FID  |
| PSLD      | 20.01 | 142.8 | 24.82 | $\underline{59.41}$ | 16.48 | $\underline{113.4}$ | 20.52 | 125.5 |
| PSLD+CFG  | 15.67 | 146.4 | 22.61 | 109.4 | 16.45 | 123.8 | 20.58 | 125.3 |
| P2L       | $\underline{21.94}$ | **72.02** | $\underline{23.02}$ | 91.15 | **19.99** | 121.7 | $\underline{20.96}$ | $\underline{85.80}$ |
| TReg      | **22.60** | $\underline{82.71}$ | **24.82** | **40.24** | $\underline{19.60}$ | **37.13** | **21.13** | **35.47** |

[Table 2]. Food - Additional data  / NFE 200
| Deblur    | | risotto → french fries |  |  | gnocchi → spaghetti  |  |
|:-:|:-:|:-:|:-:|:-:|:-:|:-:|
|           | y-MSE |  FID  | CLIP sim. | y-MSE | FID | CLIP sim. |
| PnP       | 0.029 | 339.8 | 0.278 | 0.034 | 324.4 | 0.264 |
| PSLD      | 0.015 | 249.7 | 0.254 | **0.012** | 201.7 | 0.249 |
| PSLD+CFG  | 0.015 | 268.8 | 0.264 | 0.017 | 212.2 | 0.271 |
| PSLD + PnP| 0.035 | $\underline{170.1}$ | **0.328** | 0.035 | $\underline{136.4}$ | **0.311** |
| P2L       | 0.021 | 246.8 | 0.261 | 0.016 | 203.0 | 0.254 |
| TReg      | **0.015** | **150.7** | $\underline{0.293}$ | $\underline{0.013}$ | **124.0** | $\underline{0.294}$ |


| SR    | risotto → french fries |  |  | gnocchi → spaghetti  |  |  |
|:-:|:-:|:-:|:-:|:-:|:-:|:-:|
|           | y-MSE |  FID  | CLIP sim. | y-MSE | FID | CLIP sim. |
| PnP       | **0.003** | 263.9 | 0.269 | **0.003** | 240.2 | 0.254 |
| PSLD      | 0.014 | 289.9 | 0.243 | 0.012 | 281.0 | 0.240 |
| PSLD+CFG  | 0.031 | $\underline{209.4}$ | 0.291 | 0.035 | $\underline{170.9}$ | 0.282 |
| PSLD + PnP| 0.017 | 241.0 | **0.305** | 0.015 | 232.8 | $\underline{0.286}$ |
| P2L       | 0.005 | 287.1 | 0.237 | 0.004 | 280.4 | 0.234 |
| TReg      | $\underline{0.005}$ | **170.6** | $\underline{0.296}$ | $\underline{0.004}$ | **145.4** | **0.298** |

---

### Meta-Review · Area_Chair_JULu · 2024-12-19

**Metareview:**

After rebuttal and multiple rounds of discussion, all three reviewers unanimously agreed to accept this submission，especially extra experiments over the entire dataset have been added to the rebuttal. Specifically, each reviewer's comments after rebuttal are summarized as follows.

For Reviewer CtoC, through additional experiments and detailed explanations, the authors successfully addressed all concerns raised by the reviewers, further demonstrating the effectiveness and advantages of TReg in solving inverse problems. Moreover, the authors identified promising future research directions, including the extension to natural language descriptions and the integration of more advanced text encoders.

For Reviewer vYBr, the authors effectively addressed all of the reviewers' concerns through additional experimental results and detailed explanations, thereby enhancing the completeness and academic rigor of the paper. While certain potential improvements were identified as directions for future work, the reviewers acknowledged the innovation and effectiveness of the TReg method and maintained their recommendation for the paper's acceptance.

For Reviewer 2SWh, the authors have comprehensively addressed the reviewers' main concerns through additional experiments and improvements in the manuscript's clarity, further demonstrating the applicability and innovation of TReg in tackling highly ambiguous and complex inverse problems. They have also outlined promising directions for future research, including extending the approach to more complex nonlinear problems, refining text-generated prompts, and conducting theoretical analyses of algorithmic convergence.

So,  I recommend Accept (poster).

**Additional Comments On Reviewer Discussion:**

During the rebuttal process, one reviewer mentioned that the authors' responses resolved the issues they raised in the first round of review, and raised their scores to 6, with confidence 4. Other two reviewers suggested 8 but with confidence 3.

---

### Decision · Program_Chairs · 2025-01-22

Accept (Spotlight)